# DiBS-MTL: Transformation-Invariant Multitask Learning with Direction Oracles

## Abstract

Multitask learning (MTL) algorithms typically rely on schemes that combine different task losses or their gradients through weighted averaging. These methods aim to find Pareto stationary points by using heuristics that require access to task loss values, gradients, or both. In doing so, a central challenge arises because task losses can be arbitrarily, nonaffinely scaled relative to one another, causing certain tasks to dominate training and degrade overall performance. A recent advance in cooperative bargaining theory, the Direction-based Bargaining Solution (DiBS), yields Pareto stationary solutions immune to task domination because of its invariance to monotonic nonaffine task loss transformations. However, the convergence behavior of DiBS in nonconvex MTL settings is currently not understood. To this end, we prove that under standard assumptions, a subsequence of DiBS iterates converges to a Pareto stationary point when task losses are possibly nonconvex, and propose DiBS-MTL, a computationally efficient adaptation of DiBS to the MTL setting. Finally, we validate DiBS-MTL empirically on standard MTL benchmarks, showing that it achieves competitive performance with state-of-the-art methods while maintaining robustness to nonaffine monotonic transformations that significantly degrade the performance of existing approaches, including prior bargaining-inspired MTL methods.

## 1 Introduction

The successes of deep learning have inspired investigation into "generalist" networks—models simultaneously trained for learning multiple tasks. As a result, numerous multitask learning (MTL) algorithms have been developed to tackle the inevitable conflict between task-specific loss gradients, aiming to ensure that during training, no task is under-optimized compared to others (Kendall et al., 2018; Sener & Koltun, 2018; Yu et al., 2020a; Liu et al., 2021a; Navon et al., 2022; Liu et al., 2023). However, most existing MTL methods are not robust against non-affine (monotonic) transformations to task losses, which is a crucial property desirable in the context of deep learning—where same preferences can be represented with losses of different nonaffine scalings, and it is unclear which relative scaling of the different losses ensures balanced learning without expensive and exhaustive ablations. We consider the problem of multitask learning (MTL) through the lens of centralized, cooperative bargaining methods that are invariant to *non-affine* monotonic task loss transformations.

The issue of different task losses being directly incomparable and scaled in different, non-affine fashions arises very naturally in many deep learning domains. For instance, reinforcement learning applications demand that a practitioner leverages prior, task-specific domain knowledge to design an effective reward function (Yu et al., 2025). However, in a downstream MTL setting, the loss corresponding to this reward function may dominate (or get dominated by) other task losses. At the same time, the relative performance of a "good" task-specific policy does not change when the corresponding task loss is monotonically transformed, i.e., the transformation does not change the actual underlying preferences over options. *Thus, in an MTL problem, the available task losses can be seen as monotonic—possibly non-affine—transformations of some underlying set of ideal, unknown task losses that are meaningfully scaled with respect to each other.*

Recent work in MTL has developed a connection with cooperative game theory (Navon et al., 2022). In this setting, each different loss function is a separate player in a bargaining game, and the idea is to find a balanced Pareto optimum among the players' objectives. Classical solutions to these bargain-

ing games (e.g., Nash, as explored by Navon et al. (2022)) are *not* robust to non-affine monotonic scalings, and only recently has a technique—Direction-based Bargaining Solution (`DiBS`)—been developed which remains *invariant* to these transformations (Gupta et al., 2025). However, the convergence of `DiBS` has only been analyzed in settings with strongly convex losses, and it is unclear to what extent the favorable properties of `DiBS` will apply in realistic MTL applications where task losses are almost always nonconvex. Inspired by this, we investigate the following:

1. *What theoretical properties can be established for Direction-based Bargaining Solution (`DiBS`) in the general setting where player objectives (task losses) can be non-convex?*

   **Contribution 1.** We show that under standard assumptions, for non-convex losses, a subsequence of the `DiBS` iterates provably converges to a Pareto stationary point asymptotically. Notably, our result does not require the linear independence of task loss gradients at non-Pareto stationary points, an assumption that is required by MTL methods using the Nash bargaining solution to deliver the same asymptotic guarantee (Navon et al., 2022).

2. *Can `DiBS` readily adapt to MTL applications? If so, how does it compare to existing MTL methods?*

   **Contribution 2.** We extend `DiBS` to multitask learning, showing its natural compatibility with existing bargaining-for-learning frameworks. Moreover, we propose an approximation, `DiBS-MTL`, which is computationally more efficient than `DiBS`, and also preserves desirable invariance to non-affine monotonic task loss transformations. We empirically show that `DiBS-MTL` performs competitively with existing MTL methods on widely used multitask computer vision, quantum chemistry and reinforcement learning MTL benchmarks.

   **Contribution 3.** We further investigate multitask reinforcement learning settings in which different task reward functions undergo non-affine monotonic transformations—causing potentially *non-monotonic* critic loss transformations. We empirically demonstrate that in this setting, `DiBS-MTL` maintains robust performance, while existing state-of-the-art MTL methods observe a significant drop in performance under the same transformations.

## 2 ON RELATED MTL WORKS AND EXISTING BARGAINING SOLUTIONS

### 2.1 RELATED MTL LITERATURE

The most popular MTL approach in practice is linear scalarization (`LS`)—constructing a scalarized loss by taking the unweighted sum of task losses, or using known static coefficients to compute a weighted average. While previous work has advocated for `LS` methods (Kurin et al., 2022; Xin et al., 2022), in practice, `LS` can lead to situations where certain tasks remain under-optimized. Further, it has been shown that `LS` also does not necessarily recover the entire Pareto front generated by the task losses (Hu et al., 2023). Other methods tackle the MTL problem through more sophisticated multiobjective optimization approaches, aiming to find Pareto optimal (or stationary, in general nonconvex settings) points. Such methods seek to address the task imbalances arising during training via heuristics that (i) use task-specific loss values to compute a scalar weighted average loss (but with evolving weights, unlike `LS`) (Kendall et al., 2018; Liu et al., 2019; Lin et al., 2022; Liu et al., 2023), or (ii) use task-specific loss gradients to find update directions iteratively during training (Sener & Koltun, 2018; Yu et al., 2020a; Chen et al., 2020; Liu et al., 2021a;b; Navon et al., 2022).

Existing loss-based heuristics include maximizing improvement of the worst-performing task (Liu et al., 2023), forcing improvements to be similar across tasks (Liu et al., 2019), weighting task losses randomly (Lin et al., 2022), and adapting weights according to task-based uncertainty measures (Kendall et al., 2018). Heuristics for gradient-based approaches include finding mutual task improvement directions (Yu et al., 2020a), probabilistically masking task gradients according to their sign (Chen et al., 2020), computing weights that minimize the norm of the convex combination of task gradients (Sener & Koltun, 2018), using gradient-based approaches to maximize improvement of the worst-performing task (Liu et al., 2021a), and finding the Nash bargaining solution for a bargaining subproblem (Navon et al., 2022). However, as we empirically demonstrate in Section 5, these *existing methods are not robust to monotonic, nonaffine task loss transformations*, potentially experiencing significant performance drops in such settings. We remark than one gradient-based method, `IMTL-G` (Liu et al., 2021b), in principle can produce solutions invariant to monotonic non-

affine task loss transformations. However, we show in Appendix D that even for simple convex problems, `IMTL-G` can converge to Pareto solutions that heavily favor one task over the other.

Finally, we note that other approaches also exist for the MTL problem, such as (i) task clustering, where methods first cluster tasks to reduce conflicts, and then update model training parameters for each cluster (Standley et al., 2020; Fifty et al., 2021; Song et al., 2022; Shen et al., 2024); and (ii) parameter sharing methods, which design neural network architectures consisting of task-specific and task-shared modules/parameters (Kokkinos, 2017; Guo et al., 2020; Gao et al., 2020). These approaches differ from the proposed approach at a fundamental level, in that they try to design a suite of models for the space of tasks, or design model architectures suitable for MTL, whereas the proposed approach aims to optimize a *a single* model (with a pre-defined architecture), which balances the performance of all tasks.

## 2.2 PRELIMINARIES ON COOPERATIVE BARGAINING GAMES

We provide a brief background of cooperative bargaining theory, which we utilize in our main contribution. A thorough description can be found in classical literature (Thomson, 1994; Narahari, 2014). A centralized bargaining game consists of $N$ agents and a mediator; and $\mathbf{x} \in \mathcal{S} \subseteq \mathbb{R}^n$ denotes the state of the game. We assume the $i^{\text{th}}$ agent has a differentiable cost $\ell^i(\mathbf{x}) : \mathcal{S} \to \mathbb{R}, i \in [N]$. In the context of MTL, every task can be thought of as being represented by one agent, with the agent's cost being the task loss.

Each agent want's to minimize its cost, and has preferred states $\mathbf{x}^{*,i} \in \arg\min_{\mathbf{x} \in \mathcal{S}} \ell^i(\mathbf{x})$ it wants the game to go towards. For nonconvex costs, this could be a local minimum. The goal of the mediator is to execute a bargaining strategy and find a solution state $\mathbf{x}^\dagger$. Let $\boldsymbol{\ell}(\mathbf{x}) = [\ell^i(\mathbf{x}), \ldots, \ell^N(\mathbf{x})]$. For convenience, we denote such a bargaining game by $\mathcal{B}_\mathcal{S}(\boldsymbol{\ell})$. Numerous bargaining solutions have been proposed in economics literature (Thomson, 1994), each employing different heuristics to find a Pareto optimal point—a point at which no update direction exists, which simultaneous decreases the losses for all agents. In the case when the agent losses $\ell^i(\mathbf{x})$ are nonconvex, a more appropriate goal for the mediator is to find a *Pareto stationary* point, which is a first-order necessary condition for Pareto optimality.

**Definition 1 (Pareto stationarity).** For $\mathcal{B}_\mathcal{S}(\boldsymbol{\ell})$, a point $\mathbf{x}^\dagger \in \mathcal{S}$ is Pareto stationary if $\exists \, \beta^i \geq 0, i \in [N]$, such that $\sum_{i \in [N]} \beta^i \nabla_\mathbf{x} \ell^i(\mathbf{x}^\dagger) = 0$, and $\sum_{i \in [N]} \beta^i = 1$.

We now highlight the Direction-based Bargaining Solution (`DiBS`), which we use in our method. For a bargaining game $\mathcal{B}_\mathcal{S}(\boldsymbol{\ell})$, `DiBS` finds a Pareto stationary point by taking an initial point $\mathbf{x}_1 \in \mathcal{S}$ and running the iterations

$$\mathbf{x}_{k+1} = \mathbf{x}_k - h(\mathbf{x}_k) := \mathbf{x}_k - \alpha_k \cdot \sum_{i \in [N]} \left( \|\mathbf{x}_k - \mathbf{x}^{*,i}\|_2 \cdot \frac{\nabla_\mathbf{x} \ell^i(\mathbf{x}_k)}{\|\nabla_\mathbf{x} \ell^i(\mathbf{x}_k)\|_2} \right). \tag{1}$$

**Remark 1 (Invariance of `DiBS`).** *For monotonic, possibly nonaffine transformations $h^1, \ldots, h^N$, let $\boldsymbol{h}(\boldsymbol{\ell})(\mathbf{x}) = [h^1(\ell^i)(\mathbf{x}), \ldots, h^N(\ell^N)(\mathbf{x})]$. Then `DiBS` produces the same solution for the bargaining games $\mathcal{B}_\mathcal{S}(\boldsymbol{h}(\boldsymbol{\ell}))$ and $\mathcal{B}_\mathcal{S}(\boldsymbol{\ell})$, for the same initial point $\mathbf{x}_1 \in \mathcal{S}$ and sequence of stepsizes $\{\alpha_k\}_{k \geq 0}$. This invariance to monotonic nonaffine transformations is because `DiBS` only utilizes normalized gradients and locally preferred states, i.e., local minima, both of which do not change under such transformations.*

We also note that prior has work has used the classical Nash bargaining solution (NBS) (Nash et al., 1950) in the context of MTL (Navon et al., 2022). However, the NBS is invariant to only affine agent (or task) loss transformations, and can change if the agent (or task) losses undergo monotonic *nonaffine* transformations.

As highlighted in Section 1, *the need for MTL methods which are robust to monotonic non-affine transformations makes it natural to consider incorporating `DiBS`—which is invariant to such distortions—in an MTL approach.* However, since practical MTL problems typically involve non-convex losses, it is crucial to examine the behavior of `DiBS` in the nonconvex regime, a setting not yet analyzed in prior work. This serves as our motivation for the next section.

## 3 WHAT CAN WE SAY ABOUT DiBS IN THE NONCONVEX REGIME?

In this section we establish a convergence guarantee for DiBS in the setting when agent losses can be nonconvex, which is often the case in practical MTL problems. Currently, guarantees only exist for the case when all agents have strongly convex losses, under which DiBS enjoys global asymptotic convergence to a Pareto stationary point (Gupta et al., 2025). We begin by highlighting our assumptions.

**Assumption 1.** For $\mathcal{B}_\mathcal{S}(\ell)$, the set of Pareto stationary points lies in the interior of $\mathcal{S}$, and all $\mathbf{x}^{*,i}$ exist, are finite, and are also in the interior of $\mathcal{S}$. The agent costs $\ell^i$ are differentiable, and bounded below.

**Assumption 2** (**Relaxed in Section C**). The sequence $\{\mathbf{x}_k\}_{k=1}^\infty$ generated by the DiBS iterations given in Equation (1) are bounded, i.e., there is a bounded set $\mathcal{D} \subseteq \mathcal{S}$ such that $\mathbf{x}_k \in \mathcal{D} \,\forall\, k \in \mathbb{N}$.

Assumption 1 is standard and ensures that the problem is well posed. Assumption 2 is a temporary assumption that we make for a clear exposition of our arguments while studying convergence of DiBS in nonconvex settings. Assumption 2 makes the DiBS iterates bounded, and we relax this assumption in Section C, showing that standard techniques from dynamical systems theory can be used to ensure DiBS iterates are bounded, forgoing the need for Assumption 2.

We now present our main theorem, the proof of which is in Appendix B.

**Theorem 1.** *Let $\{\mathbf{x}_k\}_{k=1}^\infty$ be the sequence generated by the DiBS algorithm given in Equation (1), for an initial point $\mathbf{x}_1 \in \mathcal{S}$ and stepsizes that follow the Robbins-Monro conditions, i.e., $\sum_k \alpha_k = \infty$, $\sum_k \alpha_k^2 < \infty$ (Robbins & Monro, 1951). Then, under Assumptions 1 and 2, the sequence $\{\mathbf{x}_k\}_{k=1}^\infty$ has a subsequence that asymptotically converges to a Pareto stationary point $\mathbf{x}^\dagger$, i.e., $h(\mathbf{x}^\dagger) = 0$.*

Theorem 1 establishes that the sequence produced by the DiBS iterates has a subsequence which converges to a Pareto stationary point, even when the agent losses are nonconvex. Note that this result is similar to a guarantee presented for the Nash bargaining solution in prior work (Navon et al., 2022). However, the existing result in prior work requires the assumption that agents' loss gradients are linearly independent at all non-Pareto stationary points. It is unclear to what extent realistic MTL problems satisfy this assumption. In comparison, our result does not require such a linear independence assumption.

## 4 ADAPTING DiBS TO MULTITASK LEARNING

We now extend DiBS to the multitask learning setting. We begin by introducing some notation, and then describe the underlying bargaining game for MTL, which has been proposed in previous work (Navon et al., 2022).

**Notation.** We use $\mathbf{0}$ to denote a zero vector of appropriate dimensions, and $\mathcal{V}(\mathbf{x}, r)$ denotes a ball of radius $r$, centered at $\mathbf{x}$. The vector $\theta \in \mathbb{R}^n$ represents shared task parameters; any additional task-specific parameters are suppressed for notational convenience and do not influence the calculation of $\theta$ iterates in this section.

**Bargaining for MTL.** The goal in MTL is to train a model to perform multiple tasks. Every learning task has an associated task loss $\ell^i(\theta) : \mathbb{R}^n \to \mathbb{R}$. During training, given parameters $\theta$, bargaining is iteratively conducted to find the next set of parameters $\theta + \Delta\theta$.

To this end, every task is represented as a distinct bargaining agent with the (minimization) objective being an approximation of the difference $\ell^i(\theta+\Delta\theta)-\ell^i(\theta)$. We consider a first-order approximation, in line with prior bargaining work for a fair comparison (Navon et al., 2022), with the $i^{\text{th}}$ agent's objective being

$$\min_{\|\Delta\theta\|_2 \leq \epsilon} \underbrace{\left(\nabla_\theta \ell^i(\theta)\right)^\top \Delta\theta}_{:=\omega^i(\Delta\theta)},$$

where we constrain the update vector $\Delta\theta$ to lie in $\mathcal{V}(\mathbf{0}, \epsilon)$, to prevent overshooting caused by large update steps. *Thus, for an MTL problem with $N$ tasks—the bargaining game becomes* $\mathcal{B}_{\mathcal{V}(\mathbf{0},\epsilon)}([\omega^1, \ldots, \omega^N])$.

**DiBS for MTL.**    We now proceed to apply `DiBS` to the bargaining game $\mathcal{B}_{\mathcal{V}(\mathbf{0},\epsilon)}([\omega^1, \ldots, \omega^N])$. Note that the bargaining game has linear objectives, and thus the normalized gradient

$$\frac{\nabla_{\Delta\theta}\omega^i}{\|\nabla_{\Delta\theta}\omega^i\|_2} = \frac{\nabla_\theta \ell^i(\theta)}{\|\nabla_\theta \ell^i(\theta)\|_2}$$

only needs to be computed once at the starting of a bargaining game. Further, the quantity $\mathbf{x}^{*,i}$ in Equation (1) has a closed form solution $\mathbf{x}^{*,i} = -\left(\nabla_\theta \ell^i(\theta)/\|\nabla_\theta \ell^i(\theta)\|_2\right) \cdot \epsilon$ for the linear objectives, i.e., the furthest point in the negative gradient direction, given that $\Delta\theta$ is constrained to lie in $\mathcal{V}(0, \epsilon)$. Thus, for an initial choice $\Delta\theta_1$ and stepsize sequence $\{\alpha_k\}$, the `DiBS` iterates for MTL become

$$\Delta\theta_{k+1} = \Delta\theta_k - \alpha_k \cdot \sum_{i\in[N]} \left(\left\|\Delta\theta_k + \epsilon \cdot \frac{\nabla_\theta \ell^i(\theta)}{\|\nabla_\theta \ell^i(\theta)\|_2}\right\|_2 \cdot \frac{\nabla_\theta \ell^i(\theta)}{\|\nabla_\theta \ell^i(\theta)\|_2}\right), k \in \mathbb{N}$$

$$\text{(Multi-step DiBS-MTL)}$$

In practice, naively applying `Multi-step DiBS-MTL` can be slow. We propose the following approximation to `DiBS`, which is more computationally efficient and still *preserves* the desirable invariance to monotonic, nonaffine task loss transformations:

$$\Delta\theta = -\epsilon \cdot \sum_{i\in[N]} \frac{\nabla_\theta \ell^i(\theta)}{\|\nabla_\theta \ell^i(\theta)\|_2} \qquad \text{(DiBS-MTL)}$$

`DiBS-MTL` can be viewed as an approximation of `Multi-step DiBS-MTL`, obtained by setting the initial $\Delta\theta_1 = 0$, running the system for a single step, i.e., only for $k = 1$, and outputting $\Delta\theta_2$ as $\Delta\theta$. Such single step approximations are common for ensuring practical computational speedups in MTL methods where an iterative processes are involved, c.f. `FAMO` (Liu et al., 2023). As we will show in Section 5, `DiBS-MTL` offers competitive performance on MTL tasks, even with a single step approximation. We remark that unlike equation 1, the single step update rule `DiBS-MTL` that does not have explicit dependencies on $\mathbf{x}^{*,i}$ due to the first-order approximation combined with the ball constraint. In Section F we provide results on a demonstrative two dimensional nonconvex example showing that the single-step version performs similarly to the multi-step version despite being relatively simpler.

## 5 EXPERIMENTS

We now evaluate `DiBS-MTL` in standard MTL benchmarks extensively used in literature (Yu et al., 2020a; Liu et al., 2021a; Navon et al., 2022; Liu et al., 2023), a demonstrative two objective example, a supervised learning computer vision and graph neural network benchmarks (`NYU-v2`, `Cityscapes`, `QM9`) (Silberman et al., 2012; Cordts et al., 2016; Blum & Reymond, 2009), and a multitask reinforcement learning dataset (Meta-World `MT10`) (Yu et al., 2020b). Our main aims are: (i) to verify that `DiBS-MTL` reliably converges to Pareto solutions from varying initializations, (ii) to compare the overall performance of solutions identified by `DiBS-MTL` to those produced by state of the art approaches across a variety of large-scale MTL benchmarks, and (iii) to verify the robustness of `DiBS-MTL` to nonaffine task loss transformations that skew the results of other methods.

### 5.1 DOES DiBS-MTL RELIABLY CONVERGE TO PARETO SOLUTIONS?

**Setting.**    We first test `DiBS-MTL` on a demonstrative two-dimensional, nonconvex multi-objective optimization example with two objectives $(\mathcal{L}_1, \mathcal{L}_2)$, which is a standard benchmark for illustrating the ability of MTL methods to achieve balanced Pareto solutions (Yu et al., 2020a; Liu et al., 2021a; Navon et al., 2022; Liu et al., 2023). The explicit forms of the objectives are provided in Appendix E.1. As shown in Figure 1, each objective in this example has deep valleys, with the bottoms having a large magnitude difference between the objective gradients, and a high (positive) curvature. Such phenomenon has been documented to exist when optimizing neural networks as well, and can lead to one objective (task) dominating others (Goodfellow et al., 2014; Yu et al., 2020a).

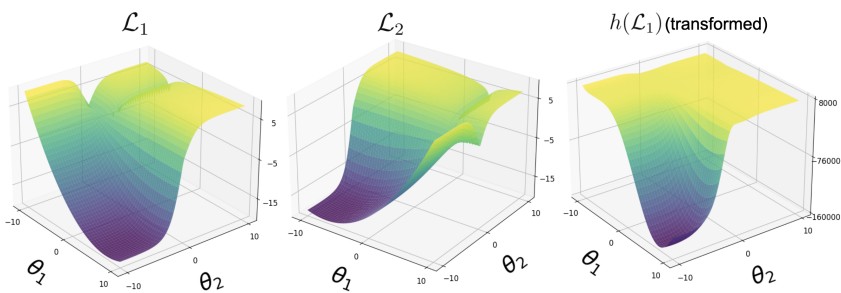

**Figure 1:** The loss functions used in the demonstrative two-dimensional example. The transformation we use is $h(\mathcal{L}) = \text{sign}(\mathcal{L}) \cdot \mathcal{L}^4$.

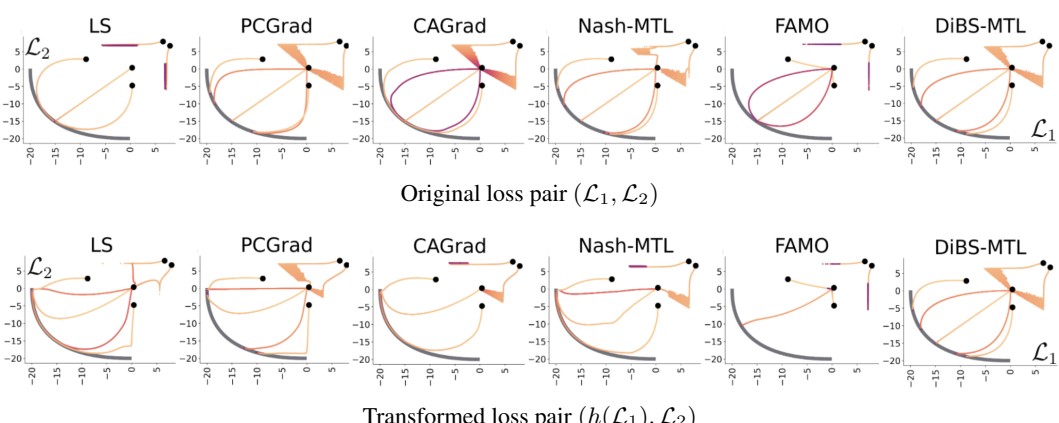

**Figure 2:** Results for the demonstrative two objective nonconvex problem. Each initialization is represented by ●, and ▬ denotes the Pareto front. We retain the original $\mathcal{L}_1$ axis in the transformed case to better visualize the different results reached by baselines. DiBS-MTL provides balanced Pareto solutions invariant to monotonic nonaffine task-loss transformations, while competing methods display degraded fairness, favoring task 1.

**Baselines.** We test DiBS-MTL along with 5 baselines—(1) linear scalarization (LS), which optimizes the unweighted loss average, (2) CAGrad (Liu et al., 2021a), (3) PCGrad (Yu et al., 2020a), (4) Nash-MTL (Navon et al., 2022), and (5) FAMO[1] (Liu et al., 2023).

**DiBS-MTL yields Pareto solutions from diverse initializations, which are unaffected by monotonic nonaffine transformations.** Figure 2 plots the solutions generated by DiBS-MTL for different initializations, for the costs $(\mathcal{L}_1, \mathcal{L}_2)$ above, as well as for $(h(\mathcal{L}_1), \mathcal{L}_2)$ where we apply the monotonic nonaffine transformation $h(x) = \text{sign}(x) \cdot x^4$ to $\mathcal{L}_1$. We observe that *DiBS-MTL consistently leads to balanced Pareto solutions for all initializations*, and these solutions remain invariant to the transformation. It is also observed that unlike DiBS-MTL, all other MTL methods produce significantly biased solutions once $\mathcal{L}_1$ is transformed (i.e., comparing the top and bottom rows of Figure 2), signaling that they are susceptible to task domination when tasks are differently scaled.

## 5.2 HOW DOES DiBS-MTL COMPARE TO EXISTING MTL METHODS?

To compare DiBS-MTL with existing methods in large-scale learning examples, we perform experiments in supervised learning (computer vision, graph neural networks) and multitask reinforcement learning (MTRL) domains that are standard benchmarks in MTL literature (Yu et al., 2020a; Liu et al., 2021a; Navon et al., 2022; Liu et al., 2023; Dai et al., 2023; Shen et al., 2024; Xiao et al.,

---

[1]We were unable to reproduce the baseline plots reported in the original FAMO paper (Liu et al., 2023). The results shown here were obtained by strictly following the default parameters and instructions in the publicly available FAMO repository https://github.com/Cranial-XIX/FAMO.

**Table 1:** Results for supervised learning `NYU-v2` benchmark. Updated baselines and DiBS variants are included. **Bold** and underline indicate best and second-best performances per column, respectively.

| | Segmentation | | Depth Estimation | | Surface Normal | | | | | | |
|---|---|---|---|---|---|---|---|---|---|---|---|
| | mIoU ↑ | Pix Acc ↑ | Abs Err ↓ | Rel Err ↓ | Median ↓ | Mean ↓ | < 30 ↑ | < 22.5 ↑ | < 11.25 ↑ | Δm%↓ | MR ↓ |
| STL | 38.30 | 63.76 | 0.68 | 0.28 | 19.21 | 25.01 | 69.15 | 57.20 | 30.14 | — | — |
| MGDA | 32.03 | 60.77 | 0.6102 | 0.2453 | **19.00** | 24.64 | **69.83** | **57.78** | **30.55** | -0.69 | 11.0 |
| UW | 39.08 | 64.73 | 0.5464 | 0.2284 | 23.04 | 27.34 | 62.85 | 49.23 | 23.49 | 3.77 | 8.2 |
| NashMTL | 40.16 | 65.65 | 0.5331 | 0.2203 | 19.96 | 25.25 | 68.29 | 55.72 | 28.62 | -3.98 | 4.2 |
| FAMO | 37.58 | 64.08 | 0.5595 | 0.2296 | 19.15 | 25.04 | 69.36 | 57.44 | 30.23 | -3.82 | 8.5 |
| LS | 40.16 | 65.63 | 0.5445 | 0.2223 | 23.03 | 27.50 | 62.66 | 49.32 | 23.61 | 3.06 | 6.0 |
| GO4Align | 38.87 | 64.52 | 0.5698 | 0.2406 | 23.41 | 27.81 | 61.83 | 48.56 | 23.45 | 5.47 | 9.9 |
| IGBv2 | 33.61 | 61.88 | 0.5728 | 0.2222 | 23.18 | 27.94 | 62.14 | 49.10 | 23.59 | 11.92 | 9.7 |
| IGBv2+Nash | 38.81 | 65.66 | **0.5258** | **0.2086** | 20.09 | 25.50 | 67.82 | 55.35 | 28.37 | 1.67 | 5.1 |
| LDC | 39.21 | 65.70 | 0.5337 | 0.2180 | 20.23 | 25.29 | 67.98 | 55.19 | 27.98 | -3.24 | 4.5 |
| DiBS-MTL | 40.92 | 66.60 | 0.5337 | 0.2216 | 20.06 | 25.35 | 68.15 | 55.54 | 28.54 | -4.11 | **3.1** |
| DiBS-MTL (5 Steps) | 41.27 | **67.10** | 0.5473 | 0.2300 | 19.69 | 25.13 | 68.69 | 56.28 | 29.31 | -4.57 | 4.2 |
| DiBS-MTL (10 Steps) | **41.33** | 67.09 | 0.5381 | 0.2260 | 19.77 | 25.06 | 68.67 | 56.12 | 29.02 | **-4.74** | 3.6 |

**Table 2:** Results on QM-9 dataset (11 tasks). Each experiment is repeated over 3 random seeds and the mean is reported. **Bold**, underlined indicate best and second-best per column.

| | $\mu$ | $\alpha$ | $\epsilon_{HOMO}$ | $\epsilon_{LUMO}$ | $\langle R^2 \rangle$ | ZPVE | $U_0$ | $U$ | $H$ | $G$ | $c_v$ | Δm%↓ | MR ↓ |
|---|---|---|---|---|---|---|---|---|---|---|---|---|---|
| | | | | | | MAE ↓ | | | | | | | |
| STL | 0.07 | 0.18 | 60.60 | 53.90 | 0.50 | 4.53 | 58.80 | 64.20 | 63.80 | 66.20 | 0.07 | — | — |
| LS | **0.0998** | 0.3108 | **69.83** | **84.87** | 5.13 | 13.43 | 137.25 | 137.97 | 138.43 | 133.55 | 0.1226 | 167.16 | 7.42 |
| GO4ALIGN | 0.1047 | 0.3175 | 70.40 | 86.48 | 5.05 | 13.35 | 138.78 | 139.49 | 139.86 | 135.27 | 0.1252 | 167.90 | 7.88 |
| UW | 0.4123 | 0.3937 | 159.07 | 145.41 | **1.09** | 4.85 | 61.52 | 61.95 | 61.98 | 61.40 | 0.1206 | 102.96 | 5.50 |
| LDC | 0.2782 | 0.3318 | 141.53 | 121.37 | 1.20 | **4.34** | **51.43** | **51.83** | **51.91** | **51.28** | 0.1076 | 68.79 | **3.50** |
| FAMO | 0.4422 | 0.3998 | 158.41 | 145.83 | 1.10 | 5.04 | 65.76 | 66.15 | 66.24 | 65.40 | 0.1267 | 110.99 | 6.67 |
| NASHMTL | 0.1024 | 0.3153 | 70.75 | 85.29 | 5.05 | 13.04 | 135.67 | 136.55 | 136.82 | 132.71 | 0.1227 | 165.12 | 6.96 |
| IGBV2 | 0.2866 | 0.3557 | 149.29 | 135.04 | 1.24 | 4.60 | 64.68 | 65.08 | 65.15 | 64.92 | 0.1176 | 85.75 | 5.33 |
| IGBV2 + Nash | 0.4399 | 0.3762 | 142.19 | 139.73 | 1.62 | 5.10 | 67.20 | 67.61 | 67.66 | 67.13 | 0.1159 | 115.24 | 7.17 |
| DiBS-MTL | 0.1347 | **0.2841** | 97.53 | 95.01 | 2.56 | 5.75 | 65.92 | 66.38 | 66.39 | 65.83 | 0.1067 | 71.83 | 4.83 |
| DiBS-MTL (5 steps) | 0.1285 | 0.2914 | 96.40 | 95.95 | 2.55 | 5.68 | 69.28 | 69.68 | 69.74 | 69.11 | 0.1065 | 72.92 | 5.54 |
| DiBS-MTL (10 steps) | 0.1213 | 0.2860 | 93.50 | 90.46 | 2.37 | 5.81 | 70.12 | 70.54 | 70.75 | 69.83 | **0.1061** | **67.37** | 5.21 |

2025). We begin by describing the setting and metrics for all datasets. We keep the default training procedures established in prior work for a fair comparison. All implementation details are given in Appendix E.

**NYU-v2 dataset.** The `NYU-v2` dataset contains 1449 aligned RGB-D indoor images (Silberman et al., 2012) with three learning tasks: (1) semantic segmentation with 13 classes, (2) continuous depth estimation, and (3) surface normal prediction. Following standard practice (Liu et al., 2019; Navon et al., 2022; Liu et al., 2023; Xiao et al., 2025; Shen et al., 2024; Dai et al., 2023), we evaluate segmentation using mean Intersection-over-Union (mIoU) and pixel accuracy; depth estimation using mean absolute error and mean relative error; and surface normals using median angular error, mean angular error, and the percentage of pixels whose angular error is below $30°$, $22.5°$, and $11.25°$ (Silberman et al., 2012). These metrics provide a comprehensive view of per-task performance across classification, regression, and geometric prediction settings.

**Cityscapes dataset.** The `Cityscapes` dataset (Cordts et al., 2016) contains 5000 street-view $128 \times 256$, RGB images annotated for two tasks: (1) semantic segmentation with 19 classes and (2) discretized depth estimation with 7 ordinal bins. Following prior MTL evaluations on Cityscapes (Navon et al., 2022; Liu et al., 2023), we report mIoU and pixel accuracy for segmentation, and absolute and relative depth error for the depth task. This dataset complements `NYU-v2` by providing a larger-scale, outdoor benchmark with visually and semantically different task relationships.

**QM9 dataset.** The `QM9` dataset Blum & Reymond (2009) is a graph neural network learning becnhmark consisting of 11 tasks, and is a widely used challenging benchmark in MTL literature (Navon et al., 2022; Liu et al., 2023; Xiao et al., 2025; Dai et al., 2023; Shen et al., 2024). It consists of

**Table 3:** Results for supervised learning `Cityscapes` benchmark. Segmentation metrics (mIoU, Pix Acc) are shown in percentage form. **Bold**, underlined indicate best and second-best per column.

| | Segmentation (%) | | Depth Estimation | | | |
|---|---|---|---|---|---|---|
| | mIoU ↑ | Pix Acc ↑ | Abs Err ↓ | Rel Err ↓ | $\Delta m\%\downarrow$ | MR ↓ |
| STL | 74.01 | 93.16 | 0.0125 | 27.77 | — | — |
| MGDA | 69.44 | 91.49 | 0.0137 | 45.26 | 20.03 | 6.3 |
| UW | 66.18 | 90.28 | 0.0158 | 54.92 | 34.41 | 10.2 |
| Nash MTL | 73.57 | 92.92 | 0.0140 | 52.34 | 25.30 | 6.0 |
| FAMO | 69.80 | 91.66 | 0.0137 | 43.66 | 18.51 | 5.5 |
| LS | 71.89 | 92.44 | 0.0180 | 111.89 | 87.64 | 10.0 |
| GO4ALIGN | 72.38 | 92.60 | 0.0312 | 93.50 | 97.26 | 9.8 |
| IGBV2 | 66.44 | 90.48 | 0.0153 | 56.56 | 34.85 | 10.0 |
| IGBV2+Nash | 73.53 | 92.95 | 0.0135 | 50.42 | 22.58 | 4.5 |
| LDC | 75.04 | 93.44 | 0.0144 | **38.44** | **12.93** | **3.0** |
| DiBS-MTL | 74.64 | 93.31 | 0.0135 | 58.09 | 29.10 | 5.9 |
| DiBS-MTL (5 Steps) | **75.46** | **93.55** | **0.0132** | 54.88 | 25.27 | 3.2 |
| DiBS-MTL (10 Steps) | 75.05 | 93.41 | 0.0133 | 52.36 | 23.34 | 3.6 |

molecules represented by annotated graphs. Our experimental setting is the same as that used in prior MTL works, and the tasks are to predict 11 different molecular properties.

**Supervised MTL baselines.** For all datasets, we compare `DiBS-MTL` with an extensive suite of strong MTL baselines: (1) linear scalarization (`LS`), (2) uncertainty weighting (`UW`) (Kendall et al., 2018), (3) `MGDA` (Sener & Koltun, 2018), (4) `Nash-MTL` (Navon et al., 2022), (5) `FAMO` (Liu et al., 2023), and four recent competitive baselines—(6) `GO4Align` (Shen et al., 2024), (7) `IGBv2`, (8) `IGBv2+Nash` (Dai et al., 2023), and (9) `LDC` (Xiao et al., 2025). All methods across all datasets share the same architecture, data pipeline, training hyperparameters and protocols, and these are taken to be the same as in prior MTL works Navon et al. (2022); Liu et al. (2023); Xiao et al. (2025); Dai et al. (2023); Shen et al. (2024); Xiao et al. (2025).

**Supervised MTL metrics.** In addition to task-specific metrics, we evaluate all methods using two standard measures of multi-task performance: (1) the mean relative per-task performance drop $\Delta m\%$ with respect to single-task learning (`STL`), and (2) the mean rank (`MR`) across all task metrics. $\Delta m\%$ captures the average improvement or degradation relative to independent single task training (`STL`), while `MR` summarizes the average rank of a method across all tasks for a benchmark. These metrics are widely used to assess whether an MTL method improves joint learning across diverse tasks (Navon et al., 2022; Liu et al., 2023; Xiao et al., 2025; Dai et al., 2023; Shen et al., 2024).

**Metaworld `MT10`.** Following prior work, we evaluate `DiBS-MTL` on the Meta-World `MT10` benchmark (Yu et al., 2020b), where a robot arm manipulator has 10 tasks, each with distinct reward functions. The complete list of tasks is given in Appendix E.

**MTRL baselines.** We compare `DiBS-MTL` with (1) `LS`, (2) `UW`, and (3) `FAMO`. We note that results for `Nash-MTL` could not be collected due to its excessive run-time for this MTRL example (c.f., Section F.1 for a runtime comparison, where `Nash-MTL` is slow even for the simpler demonstrative example). Following prior work, for every method, we use Soft Actor-Critic (SAC) (Haarnoja et al., 2018) as the underlying reinforcement learning algorithm.

**MTRL metrics.** Following prior work, we report the fraction of tasks successfully completed (Navon et al., 2022; Liu et al., 2023), where the definition of task success is as described in the Meta-World benchmark (Yu et al., 2020b). All reported results have been averaged over 10 random seeds. The implementation details are included in Appendix E.

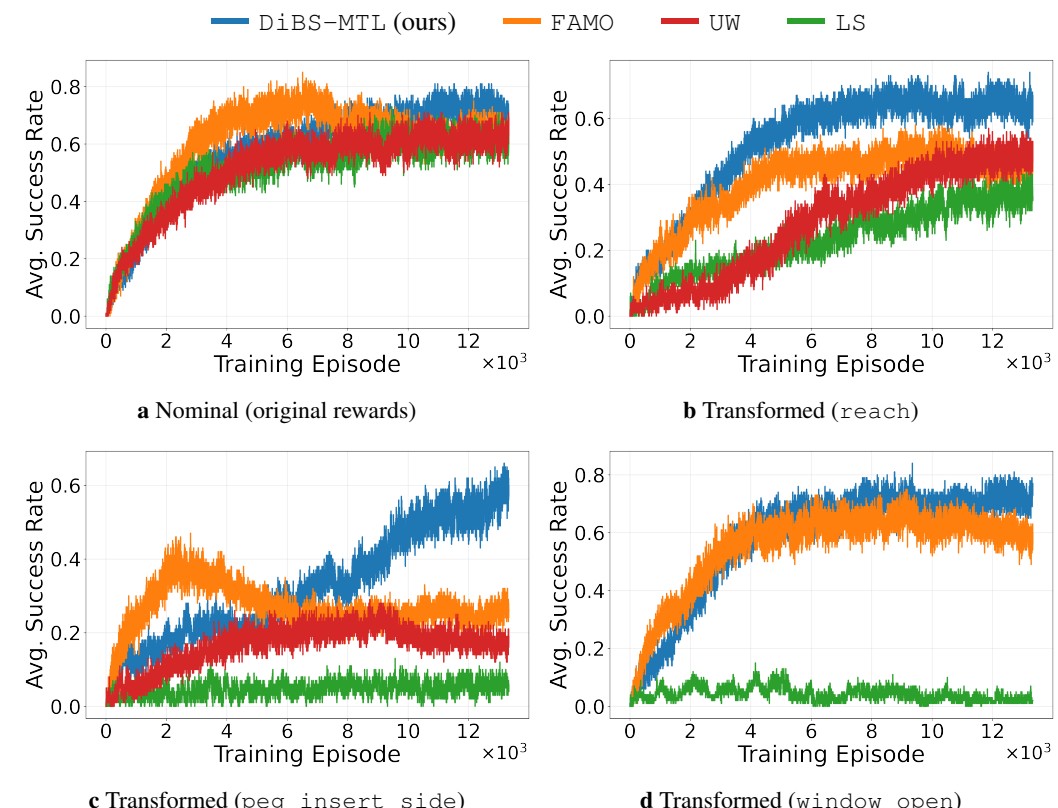

**Figure 3:** Training Curves for MTRL Meta-World `MT10` benchmark. Reward transformations significantly degrade performance of baseline MTL methods, while `DiBS-MTL` displays robustness.

**`DiBS-MTL` achieves competitive performance across all datasets.** Table 1 reports results on `NYU-v2`. `DiBS-MTL` variants *achieve state-of-the-art performance in $\Delta m\%$ as well as MR*, being the best or second-best in both MTL relevant metrics across all methods, demonstrating balanced improvements across segmentation, depth, and normal prediction. `DiBS-MTL` variants also achieve competitive or superior performance in segmentation and depth estimation, while matching strong baselines on surface normal prediction. On the `Cityscapes` benchmark, `DiBS-MTL` variants similarly perform competitively against all baselines, achieving strong segmentation performance and favorable depth metrics across the two tasks (see Table 3), and the 5 step variant of `DiBS-MTL` *achieves the second best MR*. In the `QM9` benchmark, `DiBS-MTL` variants display superior performance for the $\alpha$ and $c_v$ prediction tasks, and in the MTL relevant metrics, the 10 step `DiBS` variant *achieves the best $\Delta m$ and second best MR performance* (Table 2). Finally, in the 10 task robotic MTRL experiment as well, Table 4 and Figure 3a show that `DiBS-MTL` performs competitively, and achieves the second-best performance for the nominal case, where the reward functions for all tasks are the ones provided in Meta-World.

**Takeaway:** Across all benchmarks, `DiBS-MTL` consistently produces balanced multi-task performance and competitive performance, indicating that the `use of only normalized gradients to be invariant to monotonic non-affine transformations in DiBS-MTL does not necessarily sacrifice MTL performance.`

### 5.3 HOW ROBUST ARE `DiBS-MTL` & EXISTING MTL METHODS AGAINST NONAFFINE TRANSFORMATIONS?

We now conduct experiments in the MTRL Meta-World `MT10` benchmark Yu et al. (2020b) to investigate how `DiBS-MTL` and baseline MTL methods perform when some tasks undergo nonaffine transformations. We begin by describing the setup for the transformations. All training procedures are taken to be the same as in Section 5.2. We limit our study in this section to MTRL, because task-

**Table 4:** Best-checkpoint success (evaluated every 200 episodes) results for MTRL Meta-World `MT10` benchmark. Results are mean $\pm$ standard error over 10 seeds. **Bold** values indicate best performances per column.

| Method | Nominal | Reach | Window Open | Peg Insert |
|--------|---------|-------|-------------|------------|
| DiBS-MTL | $0.890 \pm 0.023$ | $\mathbf{0.850 \pm 0.022}$ | $\mathbf{0.920 \pm 0.025}$ | $\mathbf{0.700 \pm 0.037}$ |
| FAMO | $\mathbf{0.920 \pm 0.020}$ | $0.700 \pm 0.075$ | $0.830 \pm 0.033$ | $0.570 \pm 0.070$ |
| LS | $0.850 \pm 0.040$ | $0.480 \pm 0.044$ | $0.200 \pm 0.026$ | $0.230 \pm 0.030$ |
| UW | $0.830 \pm 0.033$ | $0.650 \pm 0.081$ | —[2] | $0.420 \pm 0.074$ |

specific losses in the supervised learning examples are relatively standardized in the literature and thus transformations there will be unwarranted (Wang et al., 2022; Azad et al., 2023). In comparison, as we mention in Section 1, in the MTRL setting, rewards for different tasks might be designed on very different scales.

**Transformed MTRL setup.** We consider transformation of 3 different tasks—for each, we apply monotonic, nonaffine transformations to the underlying reward function. For actor-critic algorithms like SAC, this transformation corresponds to potentially nonmonotonic, nonaffine critic loss transformations. This is because the losses fit the collected rewards to value functions rather than fitting the state dependent rewards. Thus, these task reward transformations can lead to potentially *non-monotonic* task loss transformations. However, we remark that in the policy landscape, a good policy for the nominal reward is also likely to perform well for the transformed reward.

**Reward transformations.** We consider the following task-transformation pairs: (i) $h(r) = \text{sign}(r) \cdot r^4$ for `reach`, (ii) $h(r) = (5 + r)^4$ for `peg insert side`, ($h$ is monotonic over the reward range), and (iii) $h(r) = \exp(r)$ for `window open`. These transformations were arbitrarily selected to illustrate a broad range of functional forms and tasks. We use the same baselines, metric, seed values, and training procedure as the nominal MTRL experiment in Section 5.2.

**`DiBS-MTL` retains performance under nonaffine reward transformations.** Figures 3a to 3d and Table 4 show that `DiBS-MTL` achieves remarkable robustness to the different reward transformations compared to existing MTL methods. `DiBS-MTL achieves state-of-the-art overall performance for all transformed cases`, while other methods face significant performance degradation in one or more cases compared to their performance in the nominal setting.

## 6 CONCLUSION

We consider the problem of constructing multitask learning (MTL) methods that are robust to monotonic, nonaffine task loss transformations. In practice, different task losses can be arbitrarily scaled with respect to one another, and those scalings can be understood as monotonic, possibly nonaffine transformations of some ideal, unknown losses which are meaningfully comparable. To address this problem, we present `DiBS-MTL`, an MTL method which is invariant to such transformations because it uses only normalized task loss gradients. Building upon recent work which formalizes the MTL training update step as a bargaining game played between tasks (Navon et al., 2022), we adapt the recently introduced `DiBS` bargaining approach (Gupta et al., 2025) in order to find a Pareto stationary (and monotonic transformation-invariant) update direction. While `DiBS` enjoys convergence guarantees when losses are strongly convex, practical MTL problems almost always have nonconvex task losses. To bridge this gap, we prove that a subsequence of the `DiBS` iterates asymptotically converges to a Pareto stationary point even in the nonconvex regime. Empirically, we demonstrate that `DiBS-MTL` achieves performance competitive with state-of-the-art methods on standard large-scale MTL benchmarks. Moreover, when task losses are nonaffinely transformed other leading MTL methods suffer degraded performance, with some tasks dominating others, whereas `DiBS-MTL` still maintains high performance, demonstrating robustness to such transformations.

---

[2]`UW` results could not be collected for the `window open` case, because the training iterations became too unstable and the underlying MuJoCo simulator (Todorov et al., 2012) consistently gave out-of-bounds errors.

## ETHICS STATEMENT

This paper presents work in the field of multitask learning. Our work has many potential applications, but we do not feel any ethical concerns need to be highlighted at this time.

## REPRODUCIBILITY STATEMENT

Implementation details for all experiments are given in Section E. Additionally, we provide all of our code in the supplementary materials. We provide setup instructions in our README, including python and package versions. The theoretical proof is given in Appendix B.

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

## A  LARGE LANGUAGE MODEL (LLM) USAGE DISCLOSURE

The authors used LLMs in the following ways.

- We used GPT-5 for spelling checks, grammar checks, and paraphrasing after the initial text is written by the authors.
- We used GPT-5 for help with understanding axis labeling options for plotting figures in the `matplotlib` package (Hunter, 2007).
- We used GPT-5 to resolve package dependency and python version issues that arose while installing the MuJoCo simulator (Todorov et al., 2012).

## B    PROOF OF THEOREM 1

**Proof sketch.**    As the iterates are bounded, there exists a subsequence which converges to some cluster point $\mathbf{x}^\dagger$. To show that $\mathbf{x}^\dagger$ is a Pareto stationary point, we will show by contradiction that $h(\mathbf{x}^\dagger) = 0$.

*Proof.* Because the `DiBS` iterates $\{\mathbf{x}_k\}_{k=1}^\infty$ is a bounded sequence from Assumption 2 (relaxed in Section C), by Bolzano-Weierstrass theorem, $\exists\, \mathbf{x}^\dagger \in \mathcal{S}$ such that a subsequence of the sequence $\{\mathbf{x}_k\}_{k=1}^\infty$ converges to $\mathbf{x}^\dagger$, and that for every neighborhood $\mathcal{U}$ of $\mathbf{x}^\dagger$, there exist infinite $n \in \mathbb{N}$ such that $\mathbf{x}_n \in \mathcal{U}$. We will show by contradiction that $h(\mathbf{x}^\dagger) = 0$. Let there exist an $a > 0$ such that $\|h(\mathbf{x}^\dagger)\|_2 = a$. We define

$$M := \sup_{\mathbf{x} \in \mathcal{D}} \|h(\mathbf{x})\|_2, \qquad u := \frac{h(\mathbf{x}^\dagger)}{\|h(\mathbf{x}^\dagger)\|_2}$$

Then, by continuity of $u^\top h(\mathbf{x})$, we have

$$\forall\, \delta > 0, \exists\, \epsilon > 0 \text{ such that } \|\mathbf{x} - \mathbf{x}^\dagger\|_2 \leq \delta \implies u^\top h(\mathbf{x}) \geq a\epsilon, \tag{2}$$

$$\text{and } \exists\, N \in \mathbb{N} \text{ such that } k \geq N \implies \alpha_k \leq \frac{a}{C} \text{ for some } C > 0. \tag{3}$$

Let $\mathcal{V}(\mathbf{x}, r)$ denote a ball in $\mathbb{R}^n$ centered at $\mathbf{x} \in \mathbb{R}^n$, with radius $r$. Let us pick a $\tilde{\delta} > 0$, and analyse the `DiBS` iterations when they are in $\mathcal{V}(\mathbf{x}^\dagger, \tilde{\delta})$. Because there an infinite number of $n$ such that $\mathbf{x}_n \in \mathcal{V}(\mathbf{x}^\dagger, \tilde{\delta})$, there are two possibilities:

1. **Case 1.** $\exists\, \delta > \tilde{\delta}$ such that the `DiBS` iterates enter $\mathcal{V}(\mathbf{x}^\dagger, \tilde{\delta})$, exit $\mathcal{V}(\mathbf{x}^\dagger, \delta)$, and then re-enter $\mathcal{V}(\mathbf{x}^\dagger, \tilde{\delta})$—an infinite number of times.

2. **Case 2.** The `DiBS` iterates enter $\mathcal{V}(\mathbf{x}^\dagger, \tilde{\delta})$ and then exit it at most a finite number of times before eventually remaining in $\mathcal{V}(\mathbf{x}^\dagger, \tilde{\delta})$ forever.

**Case 1.**    Consider the counts when the `DiBS` iterates are inside $\mathcal{V}(\mathbf{x}^\dagger, \tilde{\delta})$, $n < t_1 < t_2 < \ldots$ such that $\mathbf{x}_{t_k} \in \mathcal{V}(\mathbf{x}^\dagger, \tilde{\delta})\,\forall\, k \in \mathbb{N}$, and let the earliest corresponding counts when the `DiBS` iterates are outside $\mathcal{V}(\mathbf{x}^\dagger, \delta)$ be $e_k := \min\{t \geq t_k | \mathbf{x}_t \notin \mathcal{V}(\mathbf{x}^\dagger, \delta)\}$. Then, from Equation (2), we have

$$u^\top h(\mathbf{x}_t) \geq a\epsilon \,\forall\, t = t_k, t_{k+1}, \ldots, e_k - 1.$$

Now, from the definition of $t_k$ and $e_k$, we have

$$\|\mathbf{x}_{e_k} - \mathbf{x}_{t_k}\|_2 \geq \delta - \tilde{\delta} > 0$$

$$\implies \left\| \sum_{t=t_k}^{e_k-1} (\mathbf{x}_{t+1} - \mathbf{x}_t) \right\|_2 \geq \delta - \tilde{\delta}$$

$$\implies \sum_{t=t_k}^{e_k-1} \|\mathbf{x}_{t+1} - \mathbf{x}_t\|_2 \geq \delta - \tilde{\delta} \qquad \text{(triangle inequality)}$$

$$\implies \sum_{t=t_k}^{e_k-1} \alpha_t = \sum_{t=t_k}^{e_k-1} \frac{\|\mathbf{x}_{t+1} - \mathbf{x}_t\|_2}{\|h(\mathbf{x}_t)\|_2} \geq \sum_{t=t_k}^{e_k-1} \frac{\|\mathbf{x}_{t+1} - \mathbf{x}_t\|_2}{M} \geq \frac{\left(\delta - \tilde{\delta}\right)}{M}$$

$$\implies u^\top \left(\mathbf{x}_{e_k} - \mathbf{x}_{t_k}\right) = \sum_{t=t_k}^{e_k-1} u^\top \left(\mathbf{x}_{t+1} - \mathbf{x}_t\right) = - \sum_{t=t_k}^{e_k-1} \alpha_t u^\top h(\mathbf{x}_t)$$

$$\leq -a\epsilon \sum_{t=t_k}^{e_k-1} \alpha_t \qquad \text{(from Equation (2))}$$

$$\implies u^\top \left(\mathbf{x}_{e_k} - \mathbf{x}_{t_k}\right) \leq -\frac{a\epsilon \left(\delta - \tilde{\delta}\right)}{M}. \tag{4}$$

Here, the first equality could be legitimately written as the definition of case 1 implies that $h(\mathbf{x}_t) \neq 0, t = t_k, \ldots, e_k - 1$, otherwise the iterates would have remained within $\mathcal{V}(\mathbf{x}^\dagger, \delta)$ forever. Now, from the definition of case 1, $t_{k+1}$ and $e_k$, we have that the `DiBS` iterates cannot remain outside $\mathcal{V}(\mathbf{x}^\dagger, \tilde{\delta})$ for an infinite amount of time, and thus $\exists\, C' > 0$ such that $C' > \sum_{t=e_k}^{t_{k+1}-1} 1\ \forall\, k$. Thus

$$
u^\top \left( \mathbf{x}_{t_{k+1}} - \mathbf{x}_{e_k} \right) = \sum_{t=e_k}^{t_{k+1}-1} u^\top (\mathbf{x}_{t+1} - \mathbf{x}_t)
$$

$$
\leq \sum_{t=e_k}^{t_{k+1}-1} \alpha_t M \qquad \text{(Cauchy-Schwarz)}
$$

$$
= M \sum_{t=e_k}^{t_{k+1}-1} \alpha_t
$$

$$
\leq \frac{aMC'}{C} \qquad \text{(from Equation (3))}
$$

$$
\implies u^\top \left( \mathbf{x}_{t_{k+1}} - \mathbf{x}_{e_k} \right) \leq \frac{aMC'}{C} := a\gamma. \tag{5}
$$

Combining Equations (4) and (5), we get

$$
u^\top \left( \mathbf{x}_{t_{k+1}} - \mathbf{x}_{t_k} \right) \leq a \left( \gamma - \frac{\epsilon\left(\tilde{\delta} - \delta\right)}{M} \right)
$$

$$
\implies u^\top \mathbf{x}_{t_{k+1}} \leq u^\top \mathbf{x}_{t_1} + k \cdot a \left( \gamma - \frac{\epsilon\left(\tilde{\delta} - \delta\right)}{M} \right) \tag{6}
$$

From Cauchy-Schwarz inequality, $\|u^\top \mathbf{x}_{t_{k+1}}\|_2 \leq \|\mathbf{x}_{t_{k+1}}\|_2$ is bounded for all $k \in \mathbb{N}$ as the `DiBS` sequence is bounded. However, from Equation (6), as $k$ increases $\|u^\top \mathbf{x}_{t_{k+1}}\|_2$ becomes unbounded, which is a **contradiction**.

**Case 2.** By definition of case 2, $\exists\, t^* < \infty$ such that $\mathbf{x}_t \in \mathcal{V}(\mathbf{x}^\dagger, \delta)\ \forall\, t \geq t^*$. Thus for $T > t^*$, we have

$$
u^\top (\mathbf{x}_T - \mathbf{x}_{t^*}) = \sum_{t=t^*}^{T-1} u^\top (\mathbf{x}_{t+1} - \mathbf{x}_t)
$$

$$
= - \sum_{t=t^*}^{T-1} \alpha_t u^\top h(\mathbf{x}_t)
$$

$$
\leq -a\epsilon \sum_{t=t^*}^{T-1} \alpha_t \qquad \text{(from Equation (2))}
$$

$$
\implies u^\top \mathbf{x}_T \leq u^\top \mathbf{x}_{t^*} - a\epsilon \sum_{t=t^*}^{T-1} \alpha_t \tag{7}
$$

Similar to the argument of contradiction in case 1, $\|u^\top \mathbf{x}_{t^*}\|_2, \|u^\top \mathbf{x}_T\|_2, T \geq t^*$ are bounded. However, Equation (7) suggests that $\|u^\top \mathbf{x}_T\|$ becomes unbounded as $T$ increases, because of the Robbins-Monro stepsize condition $\sum_k \alpha_k = \infty$, and $\sum_{k=1}^{t^*-1} \alpha_k$ is finite. Hence, we arrive at a **contradiction.**

Thus, from both cases, we get that $h(\mathbf{x}^\dagger) = 0$, and from Definition 1, $\mathbf{x}^\dagger$ is a Pareto stationary point with convex coefficients

$$
\beta^i = \frac{\|\mathbf{x}^\dagger - \mathbf{x}^{*,i}\|_2 / \|\nabla_\mathbf{x} \ell^i(\mathbf{x}^\dagger)\|_2}{\sum_i \|\mathbf{x}^\dagger - \mathbf{x}^{*,i}\|_2 / \|\nabla_\mathbf{x} \ell^i(\mathbf{x}^\dagger)\|_2}.
$$

$\square$

## C  RELAXING ASSUMPTION 2

In the case when all agent (task) losses $\ell^i$ are convex, it has been established that the `DiBS` iterates are bounded (Gupta et al., 2025). In the non-convex setting, while the boundedness of the `DiBS` iterates intuitively holds for a problem with gradient conflict, i.e., task loss gradients point in opposite directions, we show that boundedness can be formally guaranteed with standard concepts from dynamical systems theory.

In particular, the following simple modification to `DiBS` allows us to guarantee boundedness, without changing the fact that the only fixed points of the dynamics are Pareto stationary points.

$$
\mathbf{x}_{k+1} = \begin{cases} f_1(\mathbf{x}_k) := \mathbf{x}_k - h(\mathbf{x}_k), & \text{if } \|\mathbf{x}_k\|_2 \leq R \text{ (standard \texttt{DiBS})} \\ f_2(\mathbf{x}_k) := \mathbf{x}_k - \alpha \frac{\mathbf{x}_k}{\|\mathbf{x}_k\|_2}, & \text{if } \|\mathbf{x}_k\|_2 \geq R + r \text{ (radially attractive)} \\ f(\mathbf{x}_k, t_k), & \text{if } R < \|\mathbf{x}_k\|_2 < R + r \text{ (convex combination)} \end{cases}
$$

where $f(\mathbf{x}_k) = (1 - g(\mathbf{x}_k)) f_1(\mathbf{x}_k) + g(\mathbf{x}_k) f_2(\mathbf{x}_k) + z(\mathbf{x}_k, t_k),$

$$
z(\mathbf{x}_k, t_k) = \left( \frac{\|\mathbf{x}_k\|_2 - R}{r} \right) \cdot \left( 1 - \frac{\|\mathbf{x}_k\|_2 - R}{r} \right) \cdot \sin(t)\boldsymbol{a}
$$

$$
g(\mathbf{x}) = \frac{e^{-\frac{r}{\|x\|_2 - R}}}{e^{-\frac{r}{\|x\|_2 - R}} + e^{-\frac{1}{1 - \frac{\|x\|_2 - R}{r}}}}, \text{ and } t_k = \sum_{i=1}^{k} \mathbb{1}\{R < \|\mathbf{x}_i\|_2 < R + r\}
$$

Here, $\boldsymbol{a} \in \mathbb{R}^n$ is any constant vector of the same dimension as the iterates $\mathbf{x}_k$. The radius $R$ can be taken to be some large positive number, larger that $\max_i \|\mathbf{x}^{*,i}\|_2$, and $r$ can be any positive constant. This modification ensures that (i) near the Pareto front, the `DiBS` iterates act as usual, (ii) in case the optimization landscape is such that due to a lack of conflicting gradient nature, the iterates somehow move away from the Pareto region and outside the $\|\mathbf{x}\|_2 \leq R + r$ ball, the iterates switch to the radially attractive dynamics and return to the ball and remain bounded, (iii) the dynamics switch is smoothly carried out in between the $\|\mathbf{x}\|_2 \leq R$ and $\|\mathbf{x}\|_2 \leq R + r$ balls, and (iv) the fixed points of the dynamics do not change— the time varying term $z(\mathbf{x}_k, t_k)$ ensures that only Pareto stationary points (all inside the $\|\mathbf{x}\|_R$ ball) are the fixed points of the modified dynamics, and the dynamics do not converge to non-Pareto stationary points.

## D  DISCUSSION ON `IMTL-G`

We elaborate on our claim made in Section 2 that though `IMTL-G` (Liu et al., 2021b) can be invariant to monotonic nonaffine transformations, it can produce Pareto solutions that are heavily favor one task over the other, possibly leading to task domination issues.

`IMTL-G` is a gradient-based method, which tries to find an update direction which has an equal projection on all task gradient vectors. Though this equal projection property brings invariance to monotonic nonaffine transformations, it also renders `IMTL-G` susceptible to task domination, as illustrated by the following simple two dimensional example.

Let $\mathcal{S} = [-1, 1] \times [-1, 1]$, with two tasks $\ell^1(x, y) = x^2 + (y - 1)^2$ and $\ell^2(x, y) = x^2 + (y + 1)^2$. Then any point of the form $(0, y), y \in [-1, 1]$ is a valid solution that `IMTL-G` can give. Even though all such points are Pareto stationary solutions, only $(0, 0)$ is balanced in the sense that it is equidistant to these symmetric functions (one function is a reflection of the other with respect to the X-axis). *Thus IMTL-G can output a point like $(0, 0.9)$ which is heavily biased towards task 1.* In contrast, even if the iterations are started at $(0, 0.9)$, `DiBS` will return the balanced point $(0, 0)$ as a solution.

# E   EXPERIMENTAL DETAILS AND CODE BASE

## E.1   DEMONSTRATIVE EXAMPLE

**Loss Functions.**   The non-convex objectives used in $(\mathcal{L}_1, \mathcal{L}_2)$ are given by

$$\mathcal{L}_1(\theta) = c_1(\theta)f_1(\theta) + c_2(\theta)g_1(\theta), \quad \mathcal{L}_2(\theta) = c_1(\theta)f_2(\theta) + c_2(\theta)g_2(\theta)$$
$$c_1(\theta) = \max\big(\tanh(0.5\,\theta_2), 0\big), \quad c_2(\theta) = \max\big(\tanh(-0.5\,\theta_2), 0\big),$$
$$f_1(\theta) = \log\Big(\max\big(|0.5(-\theta_1 - 7) - \tanh(-\theta_2)|, 10^{-6}\big)\Big) + 6,$$
$$f_2(\theta) = \log\Big(\max\big(|0.5(-\theta_1 + 3) - \tanh(-\theta_2) + 2|, 10^{-6}\big)\Big) + 6,$$
$$g_1(\theta) = \frac{(-\theta_1 + 7)^2 + 0.1(-\theta_2 - 8)^2}{10} - 20, \quad g_2(\theta) = \frac{(-\theta_1 - 7)^2 + 0.1(-\theta_2 - 8)^2}{10} - 20.$$

As mentioned in Section 5, the transformation used on $\mathcal{L}_1$ for the transformed case is the monotonic, nonaffine transformation $h(\ell) = \text{sign}(\ell) \cdot \ell^4$.

## E.2   NYUV2

**Benchmark and Setup.**   The NYU-v2 indoor scene dataset Silberman et al. (2012) provides 1,449 RGB-D images with dense per-pixel annotations for three tasks: semantic segmentation (13 classes), depth estimation, and surface normal prediction. Training uses standard task losses: per-pixel cross-entropy for segmentation, masked L1 for depth estimation over valid pixels, and a masked cosine loss over valid pixels for surface normal prediction. These choices match the NYU-v2 multitask protocol followed by prior multitask learning work (Liu et al., 2023; Navon et al., 2022). Our implementation builds on the official Nash-MTL code base[3] and incorporates the FAMO implementation[4].

**Model.**   The model architecture used in our experiments is a Multitask Attention Network (MTAN) (Liu et al., 2019) built on top of SegNet (Badrinarayanan et al., 2017). The network takes an RGB image as input and produces three task-specific outputs via separate heads: (i) a per-pixel class-score map for semantic segmentation, (ii) a scalar depth map for depth estimation, and (iii) a 3-channel surface-normal map for surface-normal prediction. For LS, MGDA, UW, FAMO, NashMTL, IGBv2, and DiBS-MTL, we use the multitask attention network (MTAN) (Liu et al., 2019) with a SegNet-style encoder–decoder architecture and apply the same training schedule, optimizer, and data-augmentation procedures as established in prior work for a fair comparison. For LDC (Xiao et al., 2025) and GO4Align (Shen et al., 2024), we use the task architectures provided in their original implementations.

**Training Protocol.**   For training, we follow the procedure outlined in (Navon et al., 2022). We train for 200 epochs with Adam (Kingma & Ba, 2014), using an initial learning rate of $1 \times 10^{-4}$ reduced to $5 \times 10^{-5}$ after 100 epochs. All reported results are averaged over three random seeds, namely $1, 7,$ and $42$.

## E.3   CITYSCAPES

**Benchmark and Setup.**   The Cityscapes outdoor driving scenes (Cordts et al., 2016) provides 5000 RGB-D images with dense per-pixel annotations for 2 tasks: semantic segmentation (7 classes) and depth estimation. Training uses standard task losses: per-pixel cross-entropy for segmentation and masked L1 for depth estimation over valid pixels These choices match the Cityscapes multitask protocol followed by prior multitask learning work (Liu et al., 2023; Navon et al., 2022).

**Model.**   Just as with NYU-v2, the model architecture used in our experiments is a Multitask Attention Network (MTAN) (Liu et al., 2019) built on top of SegNet (Badrinarayanan et al., 2017). The network takes an RGB image as input and produces two task-specific outputs via separate heads:

---

[3]https://github.com/AvivNavon/nash-mtl
[4]https://github.com/Cranial-XIX/FAMO

(i) a per-pixel class-score map for semantic segmentation, (ii) a scalar depth map for depth estimation. For `LS`, `MGDA`, `UW`, `FAMO`, `NashMTL`, `IGBv2`, and `DiBS-MTL`, we use the multitask attention network (MTAN) (Liu et al., 2019) with a SegNet-style encoder–decoder architecture and apply the same training schedule, optimizer, and data-augmentation procedures as established in prior work for a fair comparison. For `LDC` (Xiao et al., 2025) and `GO4Align` (Shen et al., 2024), we use the task architectures provided in their original implementations.

**Training Protocol.** We follow the procedure outlined in (Navon et al., 2022). We train for 300 epochs with Adam (Kingma & Ba, 2014), using an initial learning rate of $1 \times 10^{-4}$ reduced to $5 \times 10^{-5}$ after 100 epochs. All reported results are averaged over three random seeds, namely 1, 7, and 42.

### E.4 QM9

**Benchmark and Setup.** The `QM9` dataset Blum & Reymond (2009) is a graph neural network learning becnhmark consisting of 11 tasks, and is a widely used challenging benchmark in MTL literature (Navon et al., 2022; Liu et al., 2023; Xiao et al., 2025; Dai et al., 2023; Shen et al., 2024). It consists of molecules represented by annotated graphs. Our experimental setting is the same as that used in prior MTL works, and the tasks are to predict 11 different molecular properties.

**Model.** Following Navon et al. (2022), we use the PyTorch Geometric implementation of Fey & Lenssen (2019), which incorporates the widely used GNN architecture of Gilmer et al. (2017) together with the pooling operator of Vinyals et al. (2015). As with the other benchmarks, `LDC` and `GO4Align` use their own architectures.

**Training Protocol.** We train each method for 300 epochs and perform a learning-rate search based on the validation $\Delta m$ metric. A learning-rate scheduler reduces the learning rate once the validation $\Delta m$ stops improving, and the validation set is additionally used for early stopping. All reported results are averaged over three random seeds, namely 1, 7, and 42. This is in line with the approach taken by Navon et al. (2022).

### E.5 META-WORLD MT10

**Benchmark and Setup.** The MetaWorld MT10 (v1) benchmark comprises ten robotic manipulation tasks: `Reach`, `Push`, `Pick-and-Place`, `Door Open`, `Drawer Open`, `Drawer Close`, `Button Press (Top-Down)`, `Peg Insert (Side)`, `Window Open`, and `Window Close` (Yu et al., 2020b). Each task has distinct reward functions and success criteria. We build on the MTRL code base (Sodhani & Zhang, 2021)[5] with MetaWorld[6].

**Policy and Training Protocol.** We train a single Soft Actor–Critic (SAC) policy shared across all ten tasks. Multitask learning (MTL) methods are applied to balance the actor and critic updates within the shared SAC. For each MTL method, we train for 2,000,000 environment steps in total. Episodes have length 150; this corresponds to $\approx 13{,}333$ episodes overall. Evaluation is performed every 200 episodes (30,000 steps) on all tasks, and reported metrics are averaged over 10 random seeds: $1, 2, 3, 4, 5, 6, 7, 8, 9, 10$.

**Hyperparameters.** Key hyperparameter values are provided in Table 5. We follow the same hyperparemeters defined in prior works (Navon et al., 2022; Liu et al., 2023).

**Reproducibility Note.** During preliminary experiments, we observed that using the same random seed did not always produce identical training curves. This discrepancy arose because the original MTRL code (Sodhani & Zhang, 2021) did not seed the initial action sampling during the exploration phase. As a result, small differences in early exploratory actions (first 10 steps) propagated through training and produced slight variation in learning behavior even with fixed seeds.

---

[5]`https://github.com/facebookresearch/mtrl`
[6]`https://github.com/Farama-Foundation/Metaworld`

**Table 5:** Hyperparameters used for MT10 (v1) SAC.

| Component | Value |
|---|---|
| Encoder feature dim | 50 |
| Discount $\gamma$ | 0.99 |
| Initial temperature $\alpha_0$ | 1.0 |
| Actor update freq. | 1 step / env step |
| Critic target $\tau$ | 0.005 |
| Target update freq. | 1 step / env step |
| Encoder EMA $\tau_{\text{enc}}$ | 0.05 |
| Learning Rate | 0.025 |
| Update-weights cadence | every step ($= 1$) |

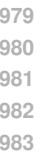

**Figure 4:** Average iterations per second in the non-transformed setting (MTRL)

## F  ADDITIONAL RESULTS

### F.1  ANALYSIS OF `DiBS-MTL` RUNTIME

In addition to examining the solutions obtained in the demonstrative example, we also evaluated the computational speed of the different MTL methods. Specifically, we compared `DiBS-MTL` with Nash-MTL and FAMO. We measured the number of iterations processed per second, with higher values indicating faster performance.

In Figure 4, we compare the runtimes of `DiBS-MTL`, FAMO, and Nash-MTL. `DiBS-MTL` runs substantially faster than Nash-MTL, achieving roughly half the speed of FAMO. This behavior is expected: unlike Nash-MTL, `DiBS-MTL` does not solve a separate optimization problem at each iteration, but it does recompute gradients and preferred states, which increases runtime relative to FAMO, which is a loss-based method and is expected to be faster.

In Table 6, we compare the runtime of different `DiBS-MTL` variants and `Nash-MTL` across the MTL benchmarks to assess how the number of bargaining steps affects computational cost. The single-step version of `DiBS-MTL` achieves the fastest runtimes on `NYU-V2` and `Cityscapes`, while falling slightly behind `Nash-MTL` on `QM9`. As the number of steps increases, the runtime grows accordingly, reflecting the added cost of performing multiple bargaining updates. At the same time, as indicated by Tables 1 to 3, single step `DiBS-MTL` achieves comparable performance to the multi-step versions.

### F.2  ADDITIONAL RESULTS IN THE DEMONSTRATIVE EXAMPLE

In addition to the results reported in section Section 5.1, we also ran additional methods in the demonstrative two-objective example. Specifically, we ran the Multiple Gradient Descent Algorithm

**Table 6:** Average Run Times in minutes for MTL Experiments averaged over 3 seeds.

| Method | NYU-V2 (200 epochs) | Cityscapes (200 epochs) | QM9 (300 epochs) |
|---|---|---|---|
| DiBS-MTL | **819.33** | **481.67** | 1705 |
| DiBS-MTL (5 Steps) | 1713 | 612 | 1719 |
| DiBS-MTL (10 Steps) | 1799 | 752 | 1726 |
| Nash-MTL | 1031 | 598 | **1699** |
| MGDA | 1047 | 494.33 | — |

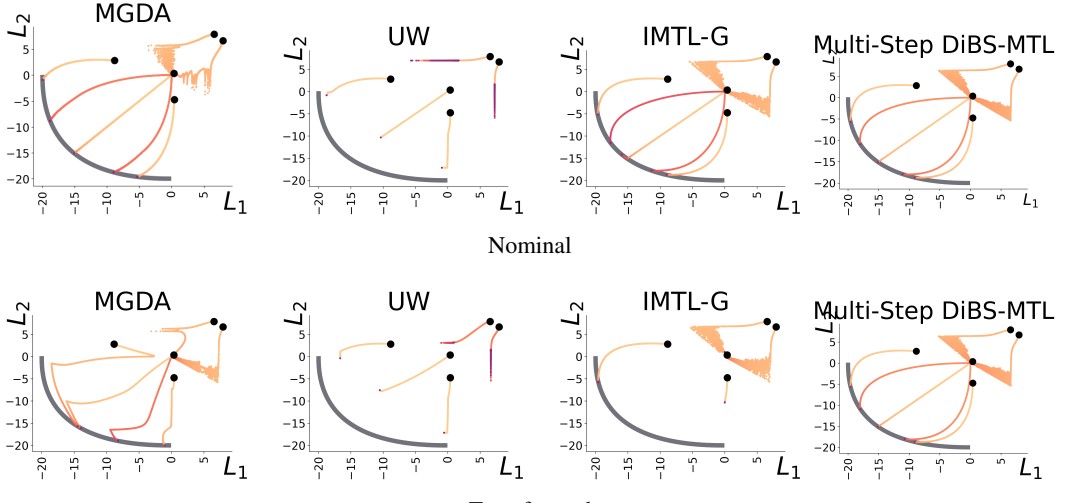

**Figure 5:** Additional experiments performed in the demonstrative two-objective example. We observe that, `Multi-step DiBS-MTL` exhibits the same invariance to monotone non-affine transforms as single-step `DiBS-MTL`, and also demonstrates similar behaviour.

(MGDA) (Désidéri, 2012), Uncertainty Weighting (UW) (Kendall et al., 2018), Impartial Multitask Learning (IMTL-G) (Liu et al., 2021b), and `Multi-step DiBS-MTL`. We run `Multi-step DiBS-MTL` for 10 steps per update. The loss functions and number of steps are identical to the experiment described in Section 5.1.

In Figure 5, we observe that the additional baseline methods of MGDA, UW, IMTL-G are not invariant to the transform, with UW failing to reach the Pareto-front in both cases. We note that `Multi-step DiBS-MTL` performs similarly to single step `DiBS-MTL`. `Multi-step DiBS-MTL` reaches the same points on the Pareto-front in both cases, showing it is also invariant to monotone non-affine transforms.

## G  ALGORITHM PSEUDOCODE

Algorithm 1 provides pseudocode for the multi-step `DiBS-MTL` algorithm. At each epoch, the algorithm approximates each agent's preferred state $\theta_i^*$ inside the approximation radius $\epsilon$: the optimal point lies at the surface of the approximation radius in the opposite direction of the agent's gradient. We then perform $S$ steps inside the approximation radius using the `DiBS` update rule for the fixed linear approximation. Finally, we update the parameter $\theta$ using the accumulated parameter difference $\Delta\theta^{(t-1)} = \theta^{(t-1,S)} - \theta^{(t-1,0)}$.

Algorithm 2 provides pseudocode for the single-step `DiBS-MTL` algorithm. When adapting `DiBS-MTL` to the single-step variant, the distances to the preferred states for each agent are equal. At $t-1 = 0$, $\|\theta^{(t-1)} - \theta_i^*\| = \epsilon$ for all $i$. This means the *approximation radius $\epsilon$ and the step size $\eta_s$ are absorbed into a single hyperparameter (i.e., learning rate $\eta$).*

---

**Algorithm 1:** $S$-step `DiBS-MTL`

---

**Input:** $\theta^{(0)}$ (initial parameters), losses $\{\ell_i\}_{i=1}^K$, learning rate $\eta$, approximation radius $\epsilon$, step size $\mu$ such that $\mu < \epsilon$, and number of epochs $T$

**for** $t = 1, \ldots, T$ **do**

    For each $i$, compute task gradients $\nabla_{\theta^{(t-1)}} \ell_i$

    For each $i$, compute preferred individual parameters inside the radius $\epsilon$:

    $\theta_i^* = \theta^{(t-1)} - \epsilon \frac{\nabla_\theta \ell^i(\theta^{(t-1)})}{\|\nabla_\theta \ell^i(\theta^{(t-1)})\|_2}$

    Set $\theta^{(t-1,0)} = \theta^{(t-1)}$

    `// DiBS updates inside the approximation radius` $\epsilon$

    **for** $s = 1, \ldots, S$ **do**

        Compute `DiBS` update direction:

        $d^{(t,s)} = \sum_{i \in [N]} \left( \|\theta^{(t-1,s-1)} - \theta_i^*\|_2 \cdot \frac{\nabla_\theta \ell^i(\theta^{(t-1,s-1)})}{\|\nabla_\theta \ell^i(\theta^{(t-1,s-1)})\|_2} \right)$

        Update $\theta^{(t-1,s)} \leftarrow \theta^{(t-1,s-1)} - \mu \, d^{(t-1,s)}$

    $\Delta \theta^{(t-1)} = \theta^{(t-1,S)} - \theta^{(t-1,0)})$

    Update $\theta^{(t)} \leftarrow \theta^{(t-1)} + \eta \Delta \theta^{(t-1)}$

**return** $\theta^{(T)}$

---

**Algorithm 2:** `DiBS-MTL` Single Step

---

**Input:** $\theta^{(0)}$ (initial parameters), losses $\{\ell_i\}_{i=1}^K$, learning rate $\eta$

**for** $t = 1, \ldots, T$ **do**

    Compute task gradients $\nabla_{\theta^{(t-1)}} \ell_i$

    Compute `DiBS` update direction: $d^{(t)} = \sum_{i \in [N]} \left( \frac{\nabla_\theta \ell^i(\theta^{(t-1)})}{\|\nabla_\theta \ell^i(\theta^{(t-1)})\|_2} \right)$

    Update $\theta^{(t)} \leftarrow \theta^{(t-1)} - \eta \, d^{(t)}$

**return** $\theta^{(T)}$

---

