# OpenReview forum: "DiBS-MTL: Transformation-Invariant Multitask Learning with Direction Oracles"
_ICLR.cc/2026/Conference — Submitted to ICLR 2026_

### Official Review · Reviewer_vr8R · 2025-10-17

**Soundness:** 2
**Presentation:** 3
**Contribution:** 2
**Rating:** 4
**Confidence:** 4

**Summary:**

This paper introduces DiBS-MTL, a new multitask learning (MTL) method based on the Direction-based Bargaining Solution (DiBS). The method is invariant to monotonic non-affine transformations of task losses, differently from previous works on MTL. The authors extend DiBS to nonconvex MTL settings and theoretically prove that a subsequence of its iterates converges to a Pareto stationary point. They then propose a practical and efficient approximation that updates parameters according to the normalized sum of task gradients. Empirical results are provided for the NYU-v2 and MT10 benchmarks. DiBS-MTL matches the performance of baselines. The paper also presents synthetic and engineered examples to demonstrate the method’s invariance to monotone transformations.

**Strengths:**

- The paper is well-written and easy to read.
- The paper provides a theoretical contribution by proving the convergence of DiBS under nonconvex losses.
- The proposed approximation (DiBS-MTL) is fairly efficient and simple, while maintaining the invariance property.

**Weaknesses:**

I generally like the paper, it is well written, theoretically grounded, and quite elegant. However, I have three main issues:
1. Approximation vs. theoretical derivation: The proposed single-step DiBS-MTL approximation appears too strong and far from the exact theoretical derivation of DiBS. It is unclear whether this approximation maintains the proven convergence properties. The paper provides no theoretical justification or formal link between the full Multi-step DiBS-MTL and its approximation, beyond an empirical comparison on a low-dimensional synthetic example. This illustrative example does not reflect realistic MTL behavior. As a result, it remains uncertain how the theoretical motivation and convergence guarantees carry over to the practical version actually used in experiments. Is it possible to provide a comparison on a larger-scale benchmark (even with a small number of DiBS steps)?
1. Motivation for invariance to monotone transformations: The motivation for focusing on invariance to monotone non-affine transformations is interesting and well-articulated, but it should be better supported with real-world evidence. Currently, the usefulness of this property is demonstrated only on synthetic and engineered examples (modified MT10), rather than on natural or widely used benchmarks where such transformations occur in practice.
1. Empirical evaluation is too limited and weak: At its current state, the empirical evidence for the effectiveness of DiBS-MTL is insufficient.
    - The paper reports results on only two datasets (NYU-v2 and MT10). I suggest adding additional common MTL benchmarks, such as QM9, Cityscapes, or CelebA, to strengthen the empirical scope.
    - The baseline coverage is limited. For NYU-v2, only NashMTL (2022) and FAMO (2023) are recent, strong baselines; others are relatively older and less strong. The authors should include more modern MTL approaches such as FairGrad, CAGrad, IMGrad, or DB-MTL. The MT10 evaluation is even more limited, with only a single recent strong baseline (which also achieves better performance compared to the proposed approach).
    - Finally, the reported results show that DiBS-MTL performs comparably to existing methods but does not significantly improve over them on any benchmark. Combined with the limited experimental coverage and limited baselines, this makes the empirical evidence inconclusive regarding the claimed advantages and effectiveness of the approach.

**Questions:**

- Could you provide a more detailed analysis of the computational complexity of DiBS-MTL, especially its scaling with the number of tasks? How does its runtime compare to recent gradient-based approaches beyond the small-scale runtime figure reported?
- The update rule depends directly on the step-size parameter $\epsilon$, but its value and tuning strategy are not reported. How was it selected in your experiments, and how sensitive is performance to this choice?

---

> ### Author Response · Authors · 2025-12-04
> **Rebuttal**
>
> We thank the reviewer for their comments. We are glad that the reviewer found our paper to be "well written", and our approach to be "theoretically grounded" and "elegant". We respond to their points in the weaknesses and questions sections point by point below.
>
> `Approximation vs. theoretical derivation:`
>
> Based on the reviewer's point, we have made an extensive revision of our experiments and conducted an empirical comparison between the approximated single-step version of DiBS-MTL with multi-step variants (5, 10 steps) for three common MTL benchmarks: NYUv2, Cityscapes, and QM9. The results (Tables 1,2,3) show that the single-step variant offers competitive performance to the multi-step variant across all benchmarks, while taking a lesser runtime (which is reported in Table 6 in the Appendix).
>
> ``Motivation for invariance to monotone transformations:``
>
> As the reviewer themselves note, "motivation for focusing on invariance to monotone non-affine transformations is interesting and well-articulated". Regarding real-world evaluation, we refrained from transforming losses for the supervised learning vision and quantum chemistry examples, as losses in these domains are very standard, and such transformations will usually be unwarranted. However, in the field of RL, such transformations can occur endogenously: the same preferences can be represented with a variety of reward functions. Our transformation functions were chosen to "stress-test" MTL methods and see how methods fare under such pronounced transformations.
>
> ``Empirical evaluation is too limited and weak:``
>
>  We have conducted a *significant expansion of our experiments* and included several new additions in the revised manuscript, which we will explain below based on the reviewer's points.
>
> * ``The paper reports results on only two datasets``
> **2 New benchmarks** Cityscapes: A standard computer-vision MTL benchmark, and QM9, a widely used graph neural network MTL benchmark.
> * ``The baseline coverage is limited.``
> **4 new recent strong baselines** from the recent past in literature - Go4Align (NeurIPS 2024) [A], igbv2 version of IGB4MTL (UAI 2023) [B], igbv2+NashMTL (best performing variant of IGB4MTL) [B], and LDC-MTL (2025) [C].
> * ``Reported results show that DiBS-MTL performs comparably to existing methods but does not significantly improve over them on any benchmark.``
> We emphasise that the aim of the experiments was to show that *the use of only normalized gradients to be invariant to monotonic non-affine transformations in DiBS-MTL does not necessarily sacrifice MTL performance.* We believe that the competitive performance of DiBS-MTL variants compared to all baselines (and many times even surpassing them on important MTL metrics) in the revised manuscript (Tables 1,2,3) demonstrates this point to a satisfying degree. Furthermore, the MT10 example demonstrates how DiBS-MTL remains robust under non-affine transformations while other methods' performances detoriate significantly.
>
>
> ``Could you provide a more detailed analysis of the computational complexity of DiBS-MTL, especially its scaling with the number of tasks? How does its runtime compare to recent gradient-based approaches beyond the small-scale runtime figure reported?``
>
> We have revised the manuscript to include runtime comparisons with gradient-based MTL methods (Table 6 in the Appendix). DiBS-MTL achieves the best runtime in NYUv2 and Cityscape benchmarks, and a very close second-best performance in QM9.
>
>
> ``The update rule depends directly on the step-size parameter, but its value and tuning strategy are not reported. How was it selected in your experiments, and how sensitive is performance to this choice?``
>
> We note that *single-step DiBS-MTL does not introduce any additional hyperparameters*, and that **the step size parameter term $\epsilon$ itself is also not a hyperparameter that requires tuning,** as it can essentially be absorbed into the learning rate. Therefore, the only tuning that single-step DiBS-MTL requires is the learning rate, which is standard in all machine learning algorithms. For multi-step DiBS MTL, the hyperparameters become $\epsilon$ and the DiBS iterations step size $\mu$ in addition to the learning rate $\eta$ (as in algorithm 1 of appendix F). We fix these to be the $1\times \eta$ and $0.1\times \eta$ the learning rate respectively.
>
> [A] Shen, Jiayi, et al. "Go4align: Group optimization for multi-task alignment." Advances in Neural Information Processing Systems 37 (2024): 111382-111405.
>
> [B] Dai, Yanqi, Nanyi Fei, and Zhiwu Lu. "Improvable gap balancing for multi-task learning." Uncertainty in Artificial Intelligence. PMLR, 2023.
>
> [C] Xiao, Peiyao, et al. "LDC-MTL: Balancing Multi-Task Learning through Scalable Loss Discrepancy Control." arXiv preprint arXiv:2502.08585 (2025).

---

### Official Review · Reviewer_Uctf · 2025-10-28

**Soundness:** 3
**Presentation:** 2
**Contribution:** 2
**Rating:** 2
**Confidence:** 3

**Summary:**

The authors propose to adapt a recently proposed method named DiBS for MTL in convex settings to general MTL problems. The main motivation for that is that DiBS is invariant to monotonic non-affine transformation, making it less susceptible to arbitrary scaling of the loss values. The authors first show a convergence guarantee to a Pareto stationary point for DiBS under standard SGD conditions. Then they apply DiBS to MTL under a bargaining formulation and present an efficient version of their method.

**Strengths:**

* Showing convergence to a Pareto stationary point asymptotically for the proposed approach under minimal and reasonable assumptions.
* Present an efficient extension of DiBS to MTL.
* Good empirical results on the NYU dataset.
* Unlike the compared approaches, the proposed approach was shown to be resilient to non-affine scalings.

**Weaknesses:**

* Regarding the method:
  - The novelty of this paper is limited. The main novel contribution of this paper, if I understand correctly, is the adaptation of DiBS to general MTL settings, namely DiBS-MTL.
  - If I understand correctly, the shared update direction is found by summation over the normalized gradients. However, while it is invariant to scaling, it loses information about the magnitude, which can be important at different stages of the learning procedure. Furthermore, it is a special case of Nash-MTL in which one assumes that all the gradients are orthogonal to each other. Perhaps I am missing something here, and I will be happy if the authors can clarify that point.
  - Related to the previous bullet, suppose we have two tasks with collinear gradients in opposite directions. The shared update direction will probably not be useful for any of them and may cause slow convergence. Taking inspiration from [1] where uncertainty was used, how does the authors suggest to handle that?
* The experimental comparisons and grounding in the MTL literature are severely lacking:
  - Missing references and *comparisons* to recent MTL studies, non-exhaustive list [1-4]. These papers should be addressed and compared to where appropriate.
  - The method was also not compared to recent and leading methods on common MTL benchmarks, such as CityScapes, QM9, and UTKFace.
  - Missing a proper comparison (on a non-syntactic benchmark) between Multi-step DiBS-MTL and DiBS-MTL to evaluate the strengths and limitations of each approach.
  - The dynamics of the loss weights across training are not clear. A proper analysis (even an empirical one) is missing.
  - Implementation details are missing. What are the hyperparameters of the method besides $\epsilon$? How sensitive is the method to these hyperparameters? Is there a special schedule for the learning rate?
* In my opinion, an algorithm of the method is missing to help better understand the update rule.

[1] [1] Achituve, I., Diamant, I., Netzer, A., Chechik, G., & Fetaya, E. (2024, July). Bayesian uncertainty for gradient aggregation in multi-task learning. In Proceedings of the 41st International Conference on Machine Learning (pp. 117-134).
[2] Senushkin, D., Patakin, N., Kuznetsov, A., & Konushin, A. (2023). Independent component alignment for multi-task learning. In Proceedings of the IEEE/CVF Conference on Computer Vision and Pattern Recognition (pp. 20083-20093).
[3] Dai, Y., Fei, N., & Lu, Z. (2023, July). Improvable gap balancing for multi-task learning. In Uncertainty in Artificial Intelligence (pp. 496-506). PMLR.
[4] Xiao, P., Dong, C., Zou, S., & Ji, K. (2025). LDC-MTL: Balancing Multi-Task Learning through Scalable Loss Discrepancy Control. arXiv preprint arXiv:2502.08585.

**Questions:**

- Regarding DiBS-MTL, it seems that $\Delta \theta$ grows with the number of tasks. How do you control its magnitude? Would it be more sensible to take the mean instead of the sum? If so, how does that affect the theory?

---

> ### Author Response · Authors · 2025-12-04
> **Rebuttal 1/2 (Method)**
>
> We thank the reviewer for their comments. We sequentially respond to the reviewer's points below.
>
> ## Regarding the method
>
> ``The novelty of this paper is limited. The main novel contribution of this paper, if I understand correctly, is the adaptation of DiBS to general MTL settings, namely DiBS-MTL.``
>  The *novelty of our paper extends beyond what the reviewer suggests*. Our paper has the following novel contributions:
>
> - We develop a **novel theoretical convergence analysis** of DiBS for **nonconvex** loss functions, which is a non-trivial advancement in the understanding of DiBS.
>
> - We develop a computationally efficient approximation of the DiBS algorithm, utilizing repeated linear approximation of losses and providing a closed-form solution for the first iterate of DiBS under this approximation.
>
> - We demonstrate the empirical performance of DiBS-MTL for a variety of MTL tasks.
>
> - We show that DiBS-MTL is also robust to non-monotone scalings of task losses.
>
>
> ``If I understand correctly, the shared update direction is found by summation over the normalized gradients. However, while it is invariant to scaling, it loses information about the magnitude, which can be important at different stages of the learning procedure.``
>
> Across all benchmarks, DiBS-MTL consistently produces balanced multi-task performance acrosss tasks and competitive performance to other baselines (see Tables 1,2,3,4 and Fig 3 in the revised manuscript), indicating that the *use of only normalized gradients to be invariant to monotonic non-affine transformations in DiBS-MTL does not necessarily sacrifice MTL performance.*
>
> Additionally, while the gradient magnitude might be informative for the learning procedure, on the other hand, *the use of magnitudes can also misinform the learning procedure as well if task losses do not share a common scaling or are adversarially scaled. As the reviewer pointed out, normalization provides robustness to such scalings.* As we state, settings with incomparable or differently scaled task losses motivate our paper, and this robustness property is a key feature for achieving MTL in such settings.
>
> ``Furthermore, it is a special case of Nash-MTL in which one assumes that all the gradients are orthogonal to each other. Perhaps I am missing something here, and I will be happy if the authors can clarify that point.``
>
> We emphasize that **DiBS-MTL is not a special case of Nash-MTL**. As we mention in the manuscript, *DiBS-MTL corresponds to a fundamentally different solution compared to Nash-MTL*. Further, our theoretical analysis does not require any orthogonality of task loss gradients. As the reviewer pointed out, *if the gradients are orthogonal to each other,* Nash-MTL outputs the average of normalized gradients. However, DiBS-MTL *does not assume* that gradients are orthogonal to each other, always utilizes the original gradient directions with no assumptions or modifications, and yields a distinct solution from Nash-MTL.
>
> ``Related to the previous bullet, suppose we have two tasks with collinear gradients in opposite directions. The shared update direction will probably not be useful for any of them and may cause slow convergence. Taking inspiration from [1] where uncertainty was used, how does the authors suggest to handle that?``
>
> The scenario described by the reviewer with the opposite gradient directions indicates that the current solution is a Pareto-optimal point. In this case, DiBS-MTL with linear approximation yields a zero magnitude update, i.e., the current point is a stationary point of DiBS-MTL.
>
> To the best of our understanding, [F] uses random task-specific weights and, consequently, random gradients, leading to tie-breaking updates. In DiBS-MTL, we use a first-order approximation for the task losses for computational advantages. On the other hand, DiBS-MTL can be easily modified to incorporate a second-order approximation for the task losses. In this case, the (unconstrained) individual solution points would not necessarily be at the same distance for each task, yielding tie-breaking updates even when two gradients point in opposing directions, favoring the underperforming tasks.
>
> References [A]-[F] are included in the part 2 of our rebuttal to the reviewer.

---

> ### Author Response · Authors · 2025-12-04
> **Rebuttal 2/2 (Experiments and other points)**
>
> ## Regarding the experiments
>  ``Missing references and comparisons to recent MTL studies``
>
> We have conducted a *significant expansion of our experiments* and included:
> - **4 new recent strong baselines** from the recent past in literature
>      Go4Align (NeurIPS 2024) [C], igbv2 version of IGB4MTL (UAI 2023) [D], igbv2+NashMTL (best performing variant of IGB4MTL) [D], and LDC-MTL (2025) [E] (these include 2 papers suggested by the reviewer).
> - **2 new widely-used, challenging benchmarks**
> Cityscapes: A standard computer-vision MTL benchmark, and QM9, a widely used graph neural network MTL benchmark.
>
>
> *In our revised manuscript, it can be seen that DiBS-MTL still performs competitively across all benchmarks (see Tables 1,2,3), with its variants frequently achieving best or second-best performance in the MTL-specific metrics across all benchmarks.*
>
> **We remark that the main strength of DiBS-MTL is to be robust to non-affine scalings while still performing competitively to other MTL methods, which do not offer such robustness. This was our main motivation, as we discuss in the paper.**
>
>
> ``Missing a proper comparison (on a non-syntactic benchmark) between Multi-step DiBS-MTL and DiBS-MTL to evaluate the strengths and limitations of each approach.``
>
>  In our revised manuscript, we have added results for multi-step DiBS-MTL (5,10 step versions) that compare performance (Tables 1,2,3) and runtimes (Table 6) across all supervised MTL benchmarks compared to single-step DiBS-MTL.
>     It can be seen that single-step DiBS-MTL offers competitive performance to multi-step DiBS-MTL variants across benchmarks while having more efficient runtimes.
>
> ``The dynamics of the loss weights across training are not clear. A proper analysis (even an empirical one) is missing.``
>
> We are not sure what the reviewer meant by "the dynamics of the loss weights". We could provide more insights if the reviewer could further eloborate on their question. Unfortunately, we are unable to provide an answer due to changes in the review process.
>
> ``Implementation details are missing. What are the hyperparameters of the method besides $\epsilon$? How sensitive is the method to these hyperparameters? Is there a special schedule for the learning rate?``
>
> We note that *single-step DiBS-MTL does not introduce any additional hyperparameters*, and that the term $\epsilon$ itself is also not a hyperparameter that requires tuning, as it can essentially be absorbed into the learning rate. Therefore, the only tuning that single-step DiBS-MTL requires is the learning rate, which is standard in all machine learning algorithms. For multi-step DiBS MTL, the hyperparameters become $\epsilon$ and the DiBS iterations step size $\mu$ in addition to the learning rate $\eta$ (as in algorithm 1 of appendix F). We fix these to be the $1\times \eta$ and $0.1\times \eta$, respectively.
>
> ``In my opinion, an algorithm of the method is missing to help better understand the update rule``
> We have added algorithms for both single and multi-step DiBS variants in Appendix F.
>
>
>
> ``Regarding DiBS-MTL, it seems that $\Delta \theta$ grows with the number of tasks. How do you control its magnitude? Would it be more sensible to take the mean instead of the sum? If so, how does that affect the theory?``
>
> We note that one only cares about the direction of $\Delta \theta$ and that the true magnitude of the step taken is determined by the algorithm's learning rate. Thus averaging or summing does not create any difference in theory as well as in practice when the learning rate is tuned. Our learning rate schedule is the same as that followed in all MTL baselines [A]-[E], and mentioned in Appendix D, following a rate of $1\times 10^{-4}$ and $1\times 10^{-5}$ for the first and second halves of the training epochs respectively for all methods.
>
> [A] Navon, Aviv, et al. "Multi-Task Learning as a Bargaining Game." International Conference on Machine Learning. PMLR, 2022.
>
> [B] Liu, Bo, et al. "Famo: Fast adaptive multitask optimization." Advances in Neural Information Processing Systems 36 (2023): 57226-57243.
>
> [C] Shen, Jiayi, et al. "Go4align: Group optimization for multi-task alignment." Advances in Neural Information Processing Systems 37 (2024): 111382-111405.
>
> [D] Dai, Yanqi, Nanyi Fei, and Zhiwu Lu. "Improvable gap balancing for multi-task learning." Uncertainty in Artificial Intelligence. PMLR, 2023.
>
> [E] Xiao, Peiyao, et al. "LDC-MTL: Balancing Multi-Task Learning through Scalable Loss Discrepancy Control." arXiv preprint arXiv:2502.08585 (2025).
>
> [F] Achituve, I., Diamant, I., Netzer, A., Chechik, G., & Fetaya, E. (2024, July). Bayesian uncertainty for gradient aggregation in multi-task learning. In Proceedings of the 41st International Conference on Machine Learning (pp. 117-134).

---

### Official Review · Reviewer_wQnK · 2025-10-31

**Soundness:** 2
**Presentation:** 2
**Contribution:** 2
**Rating:** 2
**Confidence:** 5

**Summary:**

This paper proposes DiBS-MTL, a method that computes a bargaining solution based solely on gradient direction. It effectively addresses the issue of varying loss scales, which can negatively influence the performance of current MTL methods. The paper also provides a convergence analysis in the nonconvex MTL setting, showing that the proposed method converges to a Pareto stationary point. Experiments on common MTL benchmarks demonstrate the effectiveness and robustness.

**Strengths:**

1. The comparisons of transformed loss functions are interesting (Figure 2, Figure 3); they clearly show that DiBS-MTL is more robust than other baselines.
2. Theoretical analysis is provided to demonstrate that DiBS-MTL converges to a Pareto stationary point. Additionally, extensive experiments have been conducted to show that DiBS-MTL performs comparably to existing baselines.

**Weaknesses:**

1. The theoretical analysis is based on the multi-step DiBS-MTL (Line 229-233), but in practice, the DiBS-MTL uses a single-step approximation. There should be a gap. It would be helpful if the theoretical analysis could account for the approximation error.

2. The practical DiBS-MTL reduces to the average of normalized task gradients, which is not particularly novel. The comparison between multi-step DiBS-MTL and single-step DiBS-MTL (Figure 2 vs Figure 5) is not sufficient. I would like to see more comparisons on real MTL benchmarks, such as Cityscapes, NYU-v2, and MT10, covering both performance and running time.
From the current result, it is difficult to determine whether the observed improvement comes from only gradient normalization or the DiBS framework. Additionally, it would be informative to evaluate how the baselines perform when using normalized gradients.

3. For DiBS-MTL (Line 237-240), there is a term $\epsilon$. How to set its value in practice? Are there any sensitivity analyses? Or is it absorbed into the learning rate?

4. The results shown in Table 1 are not promising enough. The baselines are too old, and many recent methods are not included.

**Questions:**

See weaknesses.

---

> ### Author Response · Authors · 2025-12-04
> **Rebuttal 1/2**
>
> We thank the reviewer for their comments. We sequentially respond to the reviewer's points below.
>
> ``The theoretical analysis is based on the multi-step DiBS-MTL (Line 229-233), but in practice, the DiBS-MTL uses a single-step approximation. There should be a gap. It would be helpful if the theoretical analysis could account for the approximation error``
> We have conducted extensive experiments to compare single and multi-step DiBS-MTL. The results show that (Tables 1,2,3 in the revised manuscript) for the same number of epochs, *performance does not significantly differ for single-step and multi-step DiBS-MTL (1,5,10 steps) variants*, while at the same time, the $k$-step DiBS-MTL variant requires $k$-times more computational effort compared to single-step DiBS-MTL for each epoch.
>
> We agree that a theoretical analysis bounding the gap between single and multi-step versions would be useful. However, conducting such an analysis is non-trivial since the distance to individual optimal points changes between steps, yielding nonuniformly weighted gradients between tasks. Consequently, we could not obtain a closed-form expression for the final point for the steps.
>
> ``The practical DiBS-MTL reduces to the average of normalized task gradients, which is not particularly novel.``
> *While **unnormalized** gradient averaging is common for MTL, we emphasize that **normalized** gradient averaging has not been explored in MTL literature to the best of our knowledge.* Further, we emphasize that the resultant unnormalized gradient averaging results from (i) using linear approximations to conduct bargaining and (ii) taking a 1-step approximation of DiBS-MTL. Both of these approximations are common in literature [A, B]. *Additionally, we respectfully want to point out that we consider the simple update rule of practical DiBS-MTL as a strength rather than a weakness, as it offers competitive performance to all baselines (as well as multi-step DiBS-MTL variants, without the extra compute effort) (Tables 1,2,3).*
>
> ``The comparison between multi-step DiBS-MTL and single-step DiBS-MTL (Figure 2 vs Figure 5) is not sufficient. I would like to see more comparisons on real MTL benchmarks, such as Cityscapes, NYU-v2, and MT10, covering both performance and running time.``
> We have revised our manuscript to include results for **2 new challenging MTL benchmarks - Cityscapes and QM9**, that compare performance and runtime for multi and single step DiBS MTL. The results (Tables 1, 2, 3) show that the single and multi-step variants show comparable performance, while the multi-step variants take significantly more time (Table 4 in the appendix).
>
> ``From the current result, it is difficult to determine whether the observed improvement comes from only gradient normalization or the DiBS framework. Additionally, it would be informative to evaluate how the baselines perform when using normalized gradients.``
>
> We would like to emphasize that the use of normalized gradients in DiBS-MTL is **not** because the task environments provide normalized gradients. DiBS-MTL uses normalized gradients as the underlying bargaining algorithm, DiBS, purposefully normalizes the task gradients. We believe that additional gradient normalization is not a suitable comparison since the gradient-based MTL methods (e.g., the baselines included in the revised manuscript) explicitly require gradient magnitudes.
>
> ``For DiBS-MTL (Line 237-240), there is a term $\epsilon$. How to set its value in practice? Are there any sensitivity analyses? Or is it absorbed into the learning rate?``
> Yes, *$\epsilon$ is absorbed into the learning rate and does not need to be tuned as an additional hyperparameter for single-step DiBS-MTL*. We note that *single-step DiBS-MTL does not introduce any additional hyperparameters*. The term $\epsilon$ itself is also not a hyperparameter that requires tuning, as it can essentially be absorbed into the learning rate. Therefore, the only tuning that single-step DiBS-MTL requires is the learning rate, which is standard in all machine learning algorithms. For multi-step DiBS MTL, the hyperparameters are the approximation radius $\epsilon$ and the DiBS iterations step size $\mu$ in addition to the learning rate $\eta$ (as in algorithm 1 of appendix F). We fix these to be the $1\times\eta$ and $0.1\times\eta$, respectively.
> Our learning rate schedule is the same as that followed in all MTL baselines [A]-[E], and mentioned in Appendix D, following a rate of $1\times 10^{-4}$ and $1\times 10^{-5}$ for the first and second halves of the training epochs, respectively, for all methods.
>
> References [A]-[E] are included in the part 2 of our rebuttal to the reviewer.

---

> ### Author Response · Authors · 2025-12-04
> **Rebuttal 2/2**
>
> ``The results shown in Table 1 are not promising enough. The baselines are too old, and many recent methods are not included.``
>
> Based on this and similar comments from other reviewers, we have conducted a *significant expansion of our experiments* and included:
> - **4 new recent strong baselines** for from the recent past in literature - Go4Align (NeurIPS 2024) [C], igbv2 version of IGB4MTL (UAI 2023) [D], igbv2+NashMTL (best performing variant of IGB4MTL) [D] and LDC-MTL (2025) [E].
> - **2 New benchmarks** Cityscapes: A standard computer-vision MTL benchmark, and QM9, a widely used graph neural network MTL benchmark.
>
>
> *In our revised manuscript it can be seen that DiBS-MTL still performs competitively across all benchmarks (see Tables 1,2,3), with its variants frequently achieving best or second-best performance in the MTL specific metrics across al benchmarks.*
>
> [A] Navon, Aviv, et al. "Multi-Task Learning as a Bargaining Game." International Conference on Machine Learning. PMLR, 2022.
>
> [B] Liu, Bo, et al. "Famo: Fast adaptive multitask optimization." Advances in Neural Information Processing Systems 36 (2023): 57226-57243.
>
> [C] Shen, Jiayi, et al. "Go4align: Group optimization for multi-task alignment." Advances in Neural Information Processing Systems 37 (2024): 111382-111405.
>
> [D] Dai, Yanqi, Nanyi Fei, and Zhiwu Lu. "Improvable gap balancing for multi-task learning." Uncertainty in Artificial Intelligence. PMLR, 2023.
>
> [E] Xiao, Peiyao, et al. "LDC-MTL: Balancing Multi-Task Learning through Scalable Loss Discrepancy Control." arXiv preprint arXiv:2502.08585 (2025).

---

### Official Review · Reviewer_Db7p · 2025-10-31

**Soundness:** 4
**Presentation:** 4
**Contribution:** 4
**Rating:** 6
**Confidence:** 3

**Summary:**

The paper presents DiBS-MTL, a method for multitask learning (MTL) designed to be robust to monotonic, non-affine transformations of task loss functions. Existing challenges in MTL arise because different task losses can be scaled differently, which may cause one task to dominate the others. DiBS-MTL uses only normalized gradients and a Direction-based Bargaining (DiBS) approach, enabling the identification of Pareto stationary points while maintaining invariance to changes in loss scaling.

Additionally, the paper shows that DiBS-MTL asymptotically converges to a Pareto stationary point even when task losses are non-convex, and it presents the mathematical mechanism of the method. The work includes experiments on two-dimensional illustrative examples to demonstrate balanced objectives, as well as applications in computer vision and multitask reinforcement learning benchmarks. Moreover, the paper details the adaptation of DiBS to existing bargaining frameworks and efficient computation of normalized gradients.

**Strengths:**

The paper introduces robustness to monotonic non-affine task loss transformations, representing a key and distinctive contribution.
The paper establishes convergence guarantees for DiBS in non-convex settings.
The paper demonstrates how DiBS can be naturally integrated with multitask learning.
The paper proposes a single-step approximation of DiBS-MTL, offering a computationally practical solution.

**Weaknesses:**

The paper tackles a relevant problem. However, the novelty of the problem itself appears somewhat incremental, which may limit the overall impact of the contribution for the community.
Some experiments are missing, as also acknowledged by the authors.

**Questions:**

Please address the weaknesses.

---

> ### Author Response · Authors · 2025-12-04
> **Rebuttal**
>
> We thank the reviewer for their comments and respond to the points they raise below.
>
> ``The novelty of the problem itself appears somewhat incremental, which may limit the overall impact of the contribution for the community``
>
> We respectfully want to point out that our paper makes some non-trivial novel contributions:
> - We develop a **novel theoretical convergence analysis** of DiBS for **nonconvex** loss functions, which is a non-trivial advancement in the understanding of DiBS. We note that the paper introducing DiBS cannot give any convergence guarantees beyond a strongly convex task loss setting [A].
> - We develop a computationally efficient approximation of the DiBS algorithm, utilizing repeated linear approximation of losses and providing a closed-form solution for the first iterate of DiBS under this approximation.
> - We demonstrate the performance of DiBS-MTL for a variety of MTL tasks.
> - We show that DiBS-MTL is also robust to non-monotone scalings of task losses.
>
>
> ``Experiments are missing:``
> We have conducted a *significant expansion of our experiments* and included:
> - **4 new recent strong baselines** from the recent past in literature - Go4Align (NeurIPS 2024) [B], igbv2 version of IGB4MTL (UAI 2023) [C], igbv2+NashMTL (best performing variant of IGB4MTL) [D], and LDC-MTL (2025) [D].
> - **2 New benchmarks** Cityscapes: A standard computer-vision MTL benchmark, and QM9, a widely used graph neural network MTL benchmark.
>
>     *In our revised manuscript, it can be seen that DiBS-MTL still performs competitively across all benchmarks (see Tables 1,2,3), with its variants frequently achieving best or second-best performance in the MTL-specific metrics across all benchmarks.*
>
> [A] Kushagra Gupta et al. (2025). Cooperative Bargaining Games Without Utilities: Mediated Solutions from Direction Oracles. In The Thirty-ninth Annual Conference on Neural Information Processing Systems (NeurIPS).
>
> [B] Shen, Jiayi, et al. "Go4align: Group optimization for multi-task alignment." Advances in Neural Information Processing Systems 37 (2024): 111382-111405.
>
> [C] Dai, Yanqi, Nanyi Fei, and Zhiwu Lu. "Improvable gap balancing for multi-task learning." Uncertainty in Artificial Intelligence. PMLR, 2023.
>
> [D] Xiao, Peiyao, et al. "LDC-MTL: Balancing Multi-Task Learning through Scalable Loss Discrepancy Control." arXiv preprint arXiv:2502.08585 (2025).

---

### Author Response · Authors · 2025-12-04
**Meta-Rebuttal: Summary of crucial points raised by reviewers**

We sincerely thank all reviewers and the area chairs for their efforts and appreciate reviewer comments that have helped us strengthen our manuscript. **In this comment, we summarize the 4 most important and common reviewer concerns and our response to them.** Edits to the manuscript reflecting this discussion are made in blue in the main text for ease of reading, and Tables 1,2,3 and 6 offer revised experiment results.


## 1) Novelty of our paper:
Several reviewers have raised points regarding the novelty of our work and the simplicity of the resultant single-step DiBS-MTL update rule. We respectfully point out that our **paper makes some non-trivial novel contributions:**
- We develop a **novel theoretical convergence analysis** of DiBS for **nonconvex** loss functions, which is a non-trivial advancement in the understanding of DiBS. We note that the paper introducing DiBS cannot give any convergence guarantees beyond a strongly convex task loss setting [A].
- We develop a computationally efficient approximation of the DiBS algorithm, utilizing repeated linear approximation of losses and providing a closed-form solution for the first iterate of DiBS under this approximation.
- We empirically show that DiBS-MTL is also robust to non-monotone scalings of task losses (apart from being invariant to monotone non-affine scalings).
- We demonstrate that DiBS-MTL has competitive performance for a variety of MTL tasks compared to strong MTL baselines, and show that the invariance to non-affine scaling of DiBS-MTL does not sacrifice MTL performance.

Further, regarding the single-step DiBS-MTL update, *while **unnormalized** gradient averaging is common for MTL, we emphasize that **normalized** gradient averaging has not been previously explored in MTL literature to the best of our knowledge.* Additionally, we respectfully want to point out that we consider **the simple update rule of practical DiBS-MTL as a strength rather than a weakness**, as it offers competitive performance to all baselines (as well as all multi-step DiBS-MTL variants, without the extra compute effort) (Tables 1,2,3).


## 2) Need of more recent baselines and more benchmarks for supervised MTL Experiments
We have conducted a *significant expansion of our experiments* and included the following in our revised manuscript:
- **4 new recent strong baselines** from the recent past in literature - Go4Align (NeurIPS 2024) [B], igbv2 version of IGB4MTL (UAI 2023) [C], igbv2+NashMTL (best performing variant of IGB4MTL) [D], and LDC-MTL (2025) [D].
- **2 New widely used challenging benchmarks** Cityscapes: A computer-vision MTL benchmark, and QM9, a graph neural network MTL benchmark.

*In our revised manuscript, it can be seen that DiBS-MTL still performs competitively across all benchmarks (see Tables 1,2,3), with its variants frequently achieving best or second-best performance in the MTL-specific metrics across all benchmarks.*

## 3) Performance differences of single and multi-step DiBS-MTL
In our revised manuscript, **we have added results for multi-step DiBS-MTL (5,10 step versions)** that compare performance (Tables 1,2,3) and runtimes (Table 6) across all supervised MTL benchmarks compared to single-step DiBS-MTL.
It can be seen that single-step DiBS-MTL offers competitive performance to multi-step DiBS-MTL variants across benchmarks while having more efficient runtimes.

## 4) Hyperparameters involved in DiBS-MTL and their values.
We note that *single-step DiBS-MTL does not introduce any additional hyperparameters besides the learning rate*. The term $\epsilon$ itself is also not a hyperparameter that requires tuning, as it can essentially be absorbed into the learning rate. *Therefore, the only tuning that single-step DiBS-MTL requires is the learning rate, which is standard in all machine learning algorithms.* For multi-step DiBS MTL, the hyperparameters become $\epsilon$ and the DiBS iterations step size $\mu$ (as in algorithm 1 of appendix F). We fix these to be the $1$ times and $0.1$ times the learning rate, respectively.





[A] Kushagra Gupta et al. (2025). Cooperative Bargaining Games Without Utilities: Mediated Solutions from Direction Oracles. In The Thirty-ninth Annual Conference on Neural Information Processing Systems (NeurIPS).

[B] Shen, Jiayi, et al. "Go4align: Group optimization for multi-task alignment." Advances in Neural Information Processing Systems 37 (2024): 111382-111405.

[C] Dai, Yanqi, Nanyi Fei, and Zhiwu Lu. "Improvable gap balancing for multi-task learning." Uncertainty in Artificial Intelligence. PMLR, 2023.

[D] Xiao, Peiyao, et al. "LDC-MTL: Balancing Multi-Task Learning through Scalable Loss Discrepancy Control." arXiv preprint arXiv:2502.08585 (2025).

---

### Meta-Review · Area_Chair_MnCw · 2025-12-29

**Summary:**

This paper studies multi-task learning (MTL) problems and aims to understand the challenges induced by varying loss scales and task conflicts. The authors propose DiBS-MTL, a method based on the Direction-based Bargaining Solution (DiBS). Unlike prior approaches, DiBS-MTL exploits only the gradient directions and therefore requires gradient normalization. This design addresses the issue of varying loss scales while maintaining invariance to loss scaling transformations. The authors further establish convergence guarantees of DiBS-MTL toward Pareto stationarity under nonconvex settings. Experimental results on several MTL benchmarks demonstrate the effectiveness of the proposed method.

**Reviewer Concerns:**

All reviewers agree that the proposed approach is interesting and that the theoretical analysis constitutes a meaningful contribution. However, several questions and concerns were raised. Reviewer Db7p questioned the novelty of the method. Reviewer wQnK raised concerns about the gap between the practical implementation, which uses a single-step update, and the theoretical analysis, which assumes multiple steps. This reviewer also questioned the novelty of the normalization idea and pointed out that the experimental evaluation is insufficient, with missing baselines and benchmark datasets. Reviewer Uctf similarly expressed concerns regarding the lack of novelty and requested comparisons with more recent baselines. Reviewer vr8R also highlighted the discrepancy between the single-step practice and the multi-step theory and raised concerns about the empirical justification of invariance to monotone transformations, as well as the limited scope of the experimental evaluation.

The authors provided detailed responses, including additional experiments on benchmark datasets such as QM9 and comparisons with methods such as LDC-MTL. After carefully reviewing the paper and the discussion, I appreciate the theoretical analysis and the idea of applying DiBS to MTL with careful performance analysis. However, I largely agree with the reviewers on the main concerns. As noted by multiple reviewers, the proposed method is closely related to Nash-MTL, with the primary distinction being the use of gradient directions while ignoring gradient magnitudes. Additional validation experiments are needed to support the theoretical claims, including real-world experiments demonstrating invariance to monotone non-affine transformations. More importantly, although the authors added further experiments, the method appears to underperform compared to some newly included baselines. Several strong gradient manipulation baselines, such as CAGrad, FairGrad, and DB-MTL, are still missing from the evaluation. In addition, many relevant works on Pareto stationarity and convergence analysis in MTL are not discussed in the current version. I strongly encourage the authors to conduct a more comprehensive review of recent advances in MTL convergence theory and to position their contributions more clearly within this literature, which would significantly strengthen the paper.

For the reasons stated above, I lean toward rejecting this paper. That said, the reviewers have provided constructive and insightful feedback, and I hope the authors will carefully address these suggestions, revise the manuscript accordingly, and consider resubmitting it to a future venue.

**Reviewer Scores:**

While the authors have made progress by including additional experiments and baselines, important methods such as CAGrad, FairGrad, and DB-MTL remain absent. I suggest that the authors consult existing MTL libraries, such as LibMTL, for implementing these baselines. Furthermore, experiments validating invariance to monotone non-affine transformations on real-world datasets are still missing. Given these remaining issues, it is less likely that the reviewers would substantially change their assessments at this stage.

---

### Decision · Program_Chairs · 2026-01-26

Reject